# Serum metabolome indicators of early childhood development in the Brazilian National Survey on Child Nutrition (ENANI-2019)

Marina Padilha[1], Victor Nahuel Keller[1], Paula Normando[1], Raquel M Schincaglia[1], Nathalia C Freitas-Costa[1], Samary SR Freire[1], Felipe M Delpino[1], Inês RR de Castro[2], Elisa MA Lacerda[3], Dayana R Farias[1], Zachary Kroezen[4], Meera Shanmuganathan[4], Philip Britz-Mckibbin[4], Gilberto Kac[1]*

[1]Department of Social and Applied Nutrition, Federal University of Rio de Janeiro, Josué de Castro Nutrition Institute, Rio de Janeiro, Brazil; [2]Department of Social Nutrition, Institute of Nutrition, State University of Rio de Janeiro, Rio de Janeiro, Brazil; [3]Department of Nutrition and Dietetics, Federal University of Rio de Janeiro, Josué de Castro Nutrition Institute, Rio de Janeiro, Brazil; [4]Department of Chemistry and Chemical Biology, McMaster University, Hamilton, Canada

**\*For correspondence:**
gilberto.kac@gmail.com

**Competing interest:** The authors declare that no competing interests exist.

## Abstract

**Background:** The role of circulating metabolites on child development is understudied. We investigated associations between children's serum metabolome and early childhood development (ECD).
**Methods:** Untargeted metabolomics was performed on serum samples of 5004 children aged 6–59 months, a subset of participants from the Brazilian National Survey on Child Nutrition (ENANI-2019). ECD was assessed using the Survey of Well-being of Young Children's milestones questionnaire. The graded response model was used to estimate developmental age. Developmental quotient (DQ) was calculated as the developmental age divided by chronological age. Partial least square regression selected metabolites with a variable importance projection ≥1. The interaction between significant metabolites and the child's age was tested.
**Results:** Twenty-eight top-ranked metabolites were included in linear regression models adjusted for the child's nutritional status, diet quality, and infant age. Cresol sulfate ($\beta$=–0.07; adjusted-p <0.001), hippuric acid ($\beta$=–0.06; adjusted-p <0.001), phenylacetylglutamine ($\beta$=–0.06; adjusted-p <0.001), and trimethylamine-*N*-oxide ($\beta$=–0.05; adjusted-p=0.002) showed inverse associations with DQ. We observed opposite directions in the association of DQ for creatinine (for children aged –1 SD: $\beta$=–0.05; p$P$=0.01;+1 SD: $\beta$=0.05; p=0.02) and methylhistidine (–1 SD: β = - 0.04; p=0.04;+1 SD: $\beta$=0.04; p=0.03).
**Conclusions:** Serum biomarkers, including dietary and microbial-derived metabolites involved in the gut-brain axis, may potentially be used to track children at risk for developmental delays.
**Funding:** Supported by the Brazilian Ministry of Health and the Brazilian National Research Council.

## Editor's evaluation

This important work advances our understanding of elements influencing neurodevelopment in children. The data presented is convincing and offers insights into the effect of demographic and environmental factors, particularly nutrition, on the functioning of the gut-brain axis and the risk for developmental delays.

## Introduction

The early years of life are characterized by remarkable growth and neurodevelopment (*UNICEF, 2017*). Child development encompasses many dimensions of a child's well-being. It is generally described into specific streams or domains of development, including motor development, speech and language progression, cognitive abilities, and socio-emotional skills (*Brown et al., 2020*).

Neurogenesis starts in the intrauterine environment and continues to shape brain morphology and plasticity after birth (*Doi et al., 2022*). The interval from birth to eight years represents a unique and critical period in which the development of a child's brain can be significantly shaped. This phenomenon encompasses special sensitivity to experiences that promote cognitive, social, emotional, and physical development (*UNICEF, 2017*). The acquisition of developmental skills results from an interplay between the development of the nervous system and other organ systems (*Brown et al., 2020*). Optimal brain development requires a stimulating environment, adequate nutrients, and social interaction with attentive caregivers (*Britto et al., 2017*).

The early childhood development (ECD) impacts long-term individual and population health outcomes, including the ability to learn, achievements in school and later life, citizenship, involvement in community activities, and overall quality of life (*van den Heuvel, 2019*). An estimated 250 million children under five in low- and middle-income countries are at risk of not attaining their developmental potential, leading to an average deficit of 19.8% in adult annual income (*Black et al., 2017*). In 2015, the importance of ECD was recognized and incorporated into the 'United Nations Sustainable Development Goals'.

Studies have demonstrated that early child metabolome disturbances may be implicated in the pathogenesis of non-typical neurodevelopment, including autism spectrum disorder (ASD; *Sotelo-Orozco et al., 2023*; *Zhu et al., 2022*), communication skills development (*Kelly et al., 2019*), and risk of impaired neurocognitive development (*Moreau et al., 2019*).

Children diagnosed with neurodevelopmental delays tend to experience more favorable treatment outcomes when these conditions are identified and addressed earlier (*Clark et al., 2018*; *Li et al., 2023*). Therefore, biomarkers are urgently needed to predict an infant's potential risk for developmental issues while gaining new insights into underlying disease mechanisms. Although child development has been a focus of research for decades, studies in low- and middle-income countries on the potential role of circulating metabolites in ECD remain limited. The present study aims to identify associations between children's serum metabolome and ECD. Identifying the relationships between metabolic phenotypes and ECD outcomes can elucidate pathways and targets for potential interventions, such as serum metabolites associated with food consumption in infancy (*Bruce et al., 2023*).

## Materials and methods

### Study design and participants

This cross-sectional study uses data from the Brazilian National Survey on Child Nutrition (ENANI-2019). ENANI-2019 is a population-based household survey with national coverage and representativeness of children aged <5 years that has investigated dietary intake, anthropometric status, and micronutrient deficiency. Details of the ENANI-2019 sample design, study completion, and methodology have been published previously (*Alves-Santos et al., 2021*; *de Castro et al., 2021*; *de Vasconcellos et al., 2021*). ENANI-2019 data collection took place from February 2019 and ended in March 2020 due to the COVID-19 pandemic.

### Covariates

Trained interviewers administered a structured questionnaire to collect socio-demographic, health and anthropometric data (*Alves-Santos et al., 2021*). The variables included in this study were: the child's age (in months), sex (male or female), educational level of the mother/caregiver of the child (0–7, 8–10, and ≥11 completed years of education), mode of delivery (vaginal or c-section), monthly familiar income (<62.2, 62.2–24.4, 124.5–248.7, >248.7 USD). Body weight (kg) and length or height (m) were used to calculate the weight for length/height z-scores (w/h z-scores). Also, the w/h z-scores were classified based on the age and sex of the child, according to World Health Organization (WHO) standards (*WHO, 2006*).

The child's diet quality was assessed using the minimum dietary diversity (MDD) indicator proposed by the WHO (WHO & United Nations Children's Fund (UNICEF), 2021). MDD requires the consumption of foods from at least five of eight food groups during the previous day. The eight food groups are (1) breast milk; (2) grains, roots, tubers, and plantains; (3) pulses (beans, peas, lentils), nuts, and seeds; (4) dairy products (milk, infant formula, yogurt, cheese); (5) flesh foods (meat, fish, poultry, organ meats); (6) eggs; (7) vitamin-A rich fruits and vegetables; and (8) other fruits and vegetables. The variable was dichotomized as children who had consumed ≥5 or<5 food groups. Data to produce this indicator was derived from the ENANI's structured questionnaire related to foods consumed the day before the first interview (*de A Lacerda et al., 2021*) Furthermore, in ENANI-2019 caregivers fulfilled one 24 hr food recall (R24h) reporting all children's food and beverage intake in the day before the interview. Child fiber intake (grams) was obtained from the R24h.

## Assessment of ECD

The Survey of Well-being of Young Children (SWYC) milestones questionnaire was used to assess ECD. This questionnaire inquiries about motor, language, and cognitive milestones appropriate for the age range of the form (*Whitesell et al., 2015*). It is recognized by the American Academy of Pediatrics and is a widely disseminated screening tool for identifying developmental delays in children aged 1–65 months (*Lipkin et al., 2020*).

The SWYC milestones questionnaire was developed and validated by *Sheldrick and Perrin, 2013*, and a version of the SWYC (SWYC-BR) has been translated, cross-culturally adapted, and validated for use in Brazilian children (*Moreira et al., 2019*). A recently published study evaluated the internal consistency of the SWYC-BR milestones questionnaire using the ENANI-2019 data and Cronbach's alpha, which showed adequate performance (0.965; 95% CI: 0.963–0.968; *Freitas-Costa et al., 2023*). SWYC-BR comprises 12 distinct forms, each aligned with the recommended age for routine pediatric wellness visits from 1 to 65 months (specifically at 1–3, 4–5, 6–8, 9–11, 12–14, 15–17, 18–22, 23–28, 29–34, 35–46, 47–58, and 59–65 months). Each form is a 10-item questionnaire. For each item, a parent/caregiver can choose one of three answers that best describe their child ('not yet', 'somewhat', or 'very much').

The ENANI-2019 data collection system automatically selected the appropriate set of developmental milestones according to the child's age. The corrected age was used to select the proper set of developmental milestones for children under two years who were born preterm (<37 gestational weeks; *Gould et al., 2021*).

## Developmental quotient

The Developmental quotient (DQ) is a continuous variable calculated by dividing developmental age by chronological age. The item response theory and graded response models were used to estimate development age (*Samejima, 1997*) according to the child's developmental milestones already reached (*Sheldrick et al., 2019*). The analysis used the full information method and incorporated the complex sample design in the Mplus software version 7 (Los Angeles, EUA; *Muthén and Muthén, 2017*). The estimated model allowed the construction of an item characteristic curve (ICC) for each milestone, representing the change in the probability of a given response (sometimes or always) and the discrimination of each milestone development by age, estimating the development age (*Freitas-Costa et al., 2023*). The ICC and its coefficients were used to estimate developmental age according to the developmental milestones reached by each child. This methodology has been previously used to assess ECD with the SWYC (*Freitas-Costa et al., 2023*; *Sheldrick et al., 2019*; *Sheldrick and Perrin, 2013*) and the Denver Test (*de L Drachler et al., 2007*).

Differently for test scores use, these methods avoid the influence of items set in the results. This approach enables the assessment of each item rather than just the final score, as the item set might be biased—meaning there could be an imbalance in the number of activities more commonly achieved among the specified items. Consequently, reaching the maximum score on the scale may be easier for certain age groups. The DQ was calculated by dividing the developmental age by the chronological age (*Freitas-Costa et al., 2023*; *Sheldrick and Perrin, 2013*). DQ equals to 1 indicates that the expected age milestones are attained. DQ values <1 or>1 suggest attaining age milestones below or above expectations. This method allows analyzing the outcome as a continuous variable.

## Blood collection

Details of the procedures adopted for blood collection and laboratory analyses have been previously described (*de Castro et al., 2021*). Fasting was not required, and changes in medication were not necessary to draw the blood sample. Briefly, 8 mL of blood sample were drawn and distributed in a trace tube (6 mL) and EDTA tube (2 mL) and transported in a cooler with a controlled temperature (from 2 °C to 8 °C) to a partner laboratory. Aliquots from the trace tube were centrifugated and the serum was transferred to a second trace tube and stored at freezing temperature (–20 °C) until laboratory analyses were performed. Serum samples with sufficient volume were stored in a biorepository (–80 °C) prior to untargeted metabolome analysis.

## Serum processing and metabolome analysis

Untargeted metabolomic analysis was performed on serum filtrate samples using a high-throughput platform based on multisegment injection-capillary electrophoresis-mass spectrometry (MSI-CE-MS). Samples were first thawed slowly on ice, where 50 µL were aliquoted and then diluted four-fold to a final volume of 200 µL in deionized water with an internal standard mix containing 40 µmol/L 3-chlorotyrosine, 3-fluorophenylalanine, 2-fluorotyrosine, trimethylamine-N-oxide[D9], γ-amino butyrate[D6], choline[D9], creatinine[D3], ornithine [$^{15}$N2], histidine[$^{15}$Nalpha], carnitine[D3], 3-methylhistidine[D3] and 2 mmol/L glucose[$^{13}$C6]. Diluted serum samples were then transferred to pre-rinsed Nanosep ultracentrifuge devices with a molecular weight cutoff of 3 kDa (Cytiva Life Sciences, Malborough, USA), and centrifuged at 10,000 × $g$ for 15 min to remove proteins. Following ultrafiltration, 20 µL of diluted serum filtrate samples were transferred to CE-compatible polypropylene vials and analyzed using MSI-CE-MS. A pooled QC was also prepared to evaluate technical precision throughput the study using 50 µL aliquots collected from the first batch of 979 serum samples processed. Overall, serum specimens were prepared and run as three separate batches of 979, 1990, and 2035 samples over an eighteen-month period. A QC-based batch correction algorithm was applied to reduce long-term system drift and improve reproducibility with QC samples analyzed in a randomized position within each analytical run (*Wehrens et al., 2016*).

High-throughput MSI-CE-MS metabolomic analyses was performed using an Agilent 6230B time-of-flight mass spectrometer (Agilent, Santa Clara, USA) with an electrospray ion source coupled to an Agilent G7100A capillary electrophoresis (CE) instrument (Agilent, Santa Clara, USA). The serum metabolome coverage comprises primarily cationic/zwitterionic and anionic polar metabolites (filtrate/unbound to protein) when using full-scan data acquisition under positive and negative ionization modes. Given the isocratic separation conditions with steady-state ionization via a sheath liquid interface, MSI-CE-MS increases sample throughput using a serial injection format where 12 samples and a pooled QC are analyzed within a single analytical run. Instrumental and data preprocessing parameters have been previously described (*Saoi et al., 2019*; *Shanmuganathan et al., 2021*).

The technical precision for serum metabolites measured in pooled QC samples had a median coefficient of variation (CV) of 10.5% with a range from 2.7 to 31% (n=422), which were analyzed by MSI-CE-MS in every analytical run throughout the study following batch correction. Overall, seventy-five circulating polar metabolites were measured in most samples (frequency >50%) with adequate technical precision (CV <30%) with the exception of symmetric dimethylarginine that was removed. Most metabolites were identified by spiking (i.e. co-migration with low mass error <5 ppm) and quantified with authentic standards, except for 13 unknown metabolites that were annotated based on their accurate mass ($m/z$), relative migration time (RMT), ionization mode (N or P), and most likely molecular formula. The metabolite distributions were severely asymmetric (average skewness = 40) and leptokurtic (average kurtosis = 1810). Therefore, a $\log_{10}$ transformation was performed on each metabolite, which reduced average skewness to 2.4 and kurtosis to 20.8. Metabolite z-scores>5 or < –5 were considered outliers and were removed (0.12% of the data).

Missing data were treated following the procedures recommended by *Wei et al., 2018* with one modification. Instead of using the "80% rule" of excluding metabolites with <80% non-missing cases (>20% missing cases) in all dependent variable categories, a less stringent 50% rule was applied to reduce the risk of excluding relevant serum metabolites. For the sole purpose of performing the exclusions, the DQ was recoded as a categorical variable (DQ ≥1 as 'within or above expectations', and DQ <1 as 'below expectations') to avoid removing metabolites that had a missingness pattern

associated with DQ. Cysteine-S-sulfate and an unknown anionic metabolite (209.030:3.04:N; $C_6H_{10}O_8$) had >50% missing cases in both DQ categories and were thus excluded.

Of the remaining 72 serum metabolites that satisfied the above selection criteria, 12.5% of the data were missing due to matrix interferences, and 1.5% were missing due to non-detection (i.e. below method detection limit). Missing data due to matrix interference were imputed using the random forest (RF) method, and non-detection missing data were imputed using quantile regression imputation of left-censored data (QRILC; *Wei et al., 2018*). The RF method used all serum metabolome data to predict what value the missing cases would likely have taken.

## Statistical analysis

We carried out descriptive and inferential analyses. The descriptive analyses were based on frequency with a 95% confidence interval (95% CI) and *Student* t-test or ANOVA were used to compare DQ in groups.

The Pearson correlation was first used to explore the correlations between circulating metabolites (exposure) and DQ (outcome). To better assess the predictiveness of each metabolite in a single model, a partial least squares regression (PLSR) was conducted (*Worley and Powers, 2013*). Partial least squares (PLS)-based analyses are the most commonly used analyses when determining the predictiveness of a large number of variables as they avoid issues with collinearity, sample size, and corrections for multiple-testing (*Blekherman et al., 2011*; *Wold et al., 2001*; *Gromski et al., 2015*). The PLSR reduces the metabolites to orthogonal components, which are maximally predictive of the outcome and generate an indicator of how much each metabolite contributes to predicting the outcome, called the variable importance projection (VIP). Because our goal was not to determine the components that are maximally predictive of DQ but to rank the metabolites on their contribution to predicting the outcome, we focused on the VIPs from this analysis.

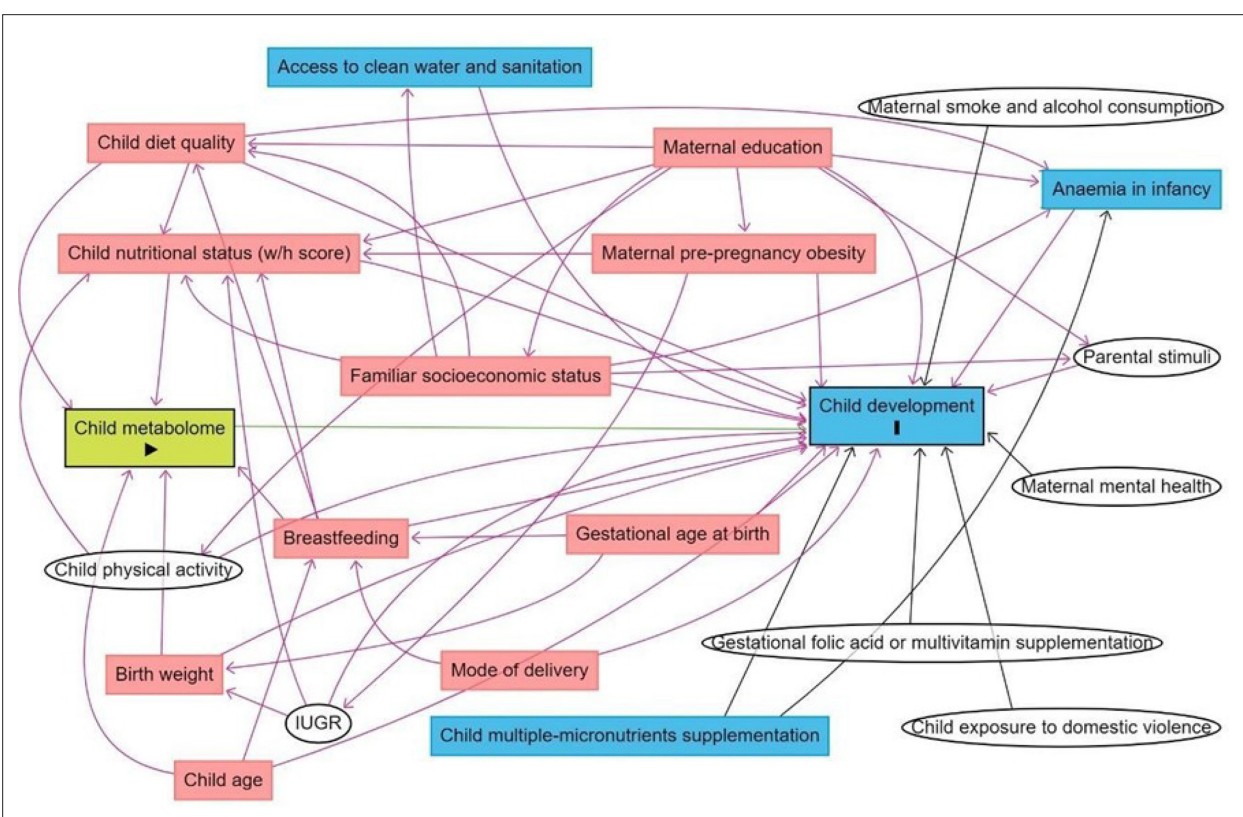

**Figure 1.** Directed acyclic graph (DAG) of the association between child serum metabolome and early childhood development. Note: IUGR: intrauterine growth restriction. Breastfeeding refers to breastfeeding practice exclusively until 6 months and/or complemented until 2 years. Minimum adjustments suggested by the DAG: Birth weight, breastfeeding, child age, child diet quality, child nutritional status (w/h z-score). Blue rectangles: ancestor of the outcome; pink rectangles: ancestor of exposure and outcome; white circles: non observed variables in the ENANI-2019.

The PLSR was trained on 80% of the data, and the remaining 20% was used as test data. Training and test data were randomly allocated. The model with the optimal number of components considering predictive value and parsimony was used to generate VIP values. Serum metabolites were selected for further analyses if they had a VIP ≥1.

The subsequent step was to disentangle the selected metabolites from confounding variables. A Directed Acyclic Graph (DAG; *Breitling et al., 2022*) was used to more objectively determine the minimally sufficient adjustments for the regression models to account for potentially confounding variables while avoiding collider variables and variables in the metabolite-DQ causal pathways, which if controlled for would unnecessarily remove explained variance from the metabolites and hamper our ability to detect biomarkers. To minimize bias from subjective judgments of which variables should and should not be included as covariates, the DAG only included variables for which there was evidence from systematic reviews or meta-analysis of relationships with both the metabolome and DQ (*Figure 1*). Birth weight, breastfeeding, child's diet quality, the child's nutritional status, and the child's age were the minimal adjustments suggested by the DAG. Birth weight was a variable with high missing data, and indicators of breastfeeding practice data (referring to exclusive breastfeeding until 6 months and/or complemented until 2 years) were collected only for children aged 0–23 months. Therefore, those confounders were not included as adjustments. Child's diet quality was evaluated as MDD, the child's nutritional status as w/h z-score, and the child's age in months.

Multiple linear regression between each metabolite and DQ were performed and adjusted for the covariates. Additional regressions were done to explore interactions between the metabolites, sex, and age.

Since several circulating metabolites most associated with DQ are relevant to microbiome health, those circulating metabolites may be biomarkers of a beneficial effect of the microbiome on development. We employed mediation analyses to explore the potential role of specific serum metabolites as mediators in the relationship between certain exposure variables related to the microbiome establishment in early life, such as mode of delivery (*Reyman et al., 2019*), child's diet quality (*Baldeon et al., 2023*), as well as child fiber intake (*Cronin et al., 2021*) and DQ. For the mediation analyses, we adopted the approach proposed by *Tingley et al., 2014*, which provides independent estimates for the average causal mediation effect (ACME - the effect of the exposure variable on DQ that is mediated by the metabolite), the average direct effect (ADE - controlling for metabolite concentrations) of the exposure variable on DQ, and the total effect of the exposure variable on DQ (mediation plus direct effect). Bootstrap tests using 5,000 iterations evaluated whether the effects were statistically significant. Due to the exploratory nature of the mediation analysis, significance was not corrected for multiple testing. The child's age (in months) and w/h z-score were entered as covariates.

All other results were considered statistically significant at an adjusted-p ≤0.05 after the Benjamini-Hochberg correction for multiple comparisons. Statistical analyses were carried out using the R programming language (R Core Team; http://www.R-project.org), through JupyterLab, using the following packages: ggplot2 (http://ggplot2.org), interactions (https://cran.r-project.org), dplyr (https://cran.r-project.org), tidyverse (https://www.tidyverse.org/), pls (*Mevik and Wehrens, 2007*), plsVarSel (https://github.com/khliland/plsVarSel), mediation (*Tingley et al., 2014*).

## Ethical aspects

The ENANI-2019 was approved by the Research Ethics Committee of the Clementino Fraga Filho University Hospital of the Federal University of Rio de Janeiro (UFRJ) under the number CAAE 89798718.7.0000.5257. Data were collected after a parent/caregiver of the child authorized participation in the study through an informed consent form and following the principles of the Declaration of Helsinki.

## Results

In total, 14,558 children under five years were evaluated, and 12,598 children aged 6–59 months were eligible for blood collection, of whom 8829 (70%) had the biological material collected. Due to the costs involved in the metabolome analysis, it was necessary to reduce the sample size that is equivalent to 57% of total participants from ENANI-2019 with stored blood specimens. Therefore, the infants were stratified by age groups (6–11, 12–23, and 24–59 months) and health conditions such

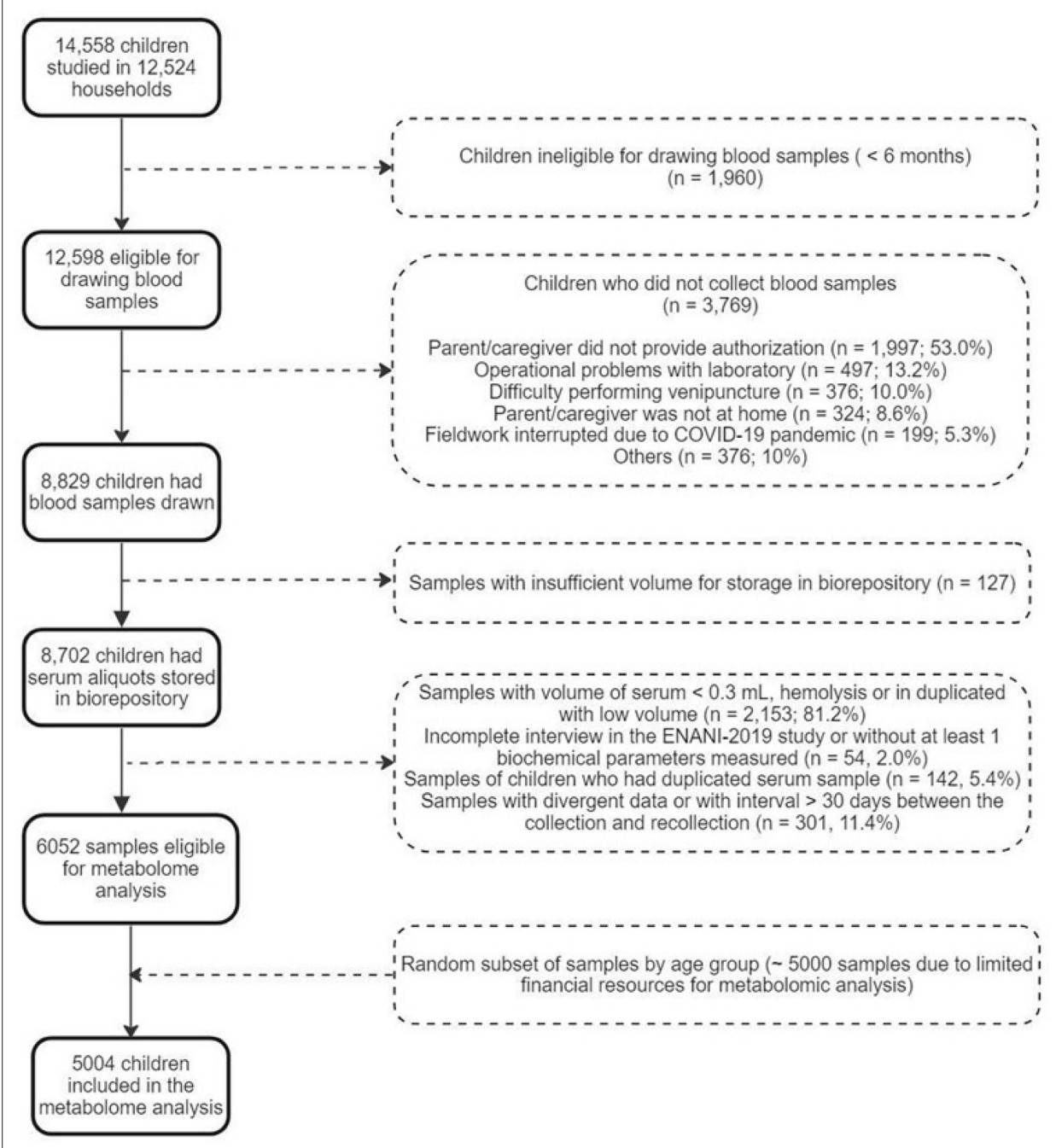

**Figure 2.** Flow chart of children included in the serum metabolome analysis of ENANI-2019 (the Brazilian National Survey on Child Nutrition). Note: COVID-19, coronavirus disease 2019.

as anemia and micronutrient deficiencies. The selection process aimed to represent diverse health statuses to the original sample. Ultimately, 5004 children were selected for the final sample through a random sampling process that ensured a balanced representation across these groups (*Figure 2*).

The mean infant age was 34 months, and 48.9% of the participants were between 36 and 59 months. Almost half of the children evaluated lived in the North (24.8%) and Northeast (22.6%) regions. Most children were of normal weight, and 25% were at risk or had excessive weight. The prevalence of MDD was 59.3%. Most children (72.4%) lived in households with a monthly income greater than USD 248.7, the Brazilian minimum wage, and 51% had a mother/caregiver with at least 11 years of education (*Table 1*).

**Table 1.** Characteristics of children 6–59 months evaluated in a subset sample of ENANI-2019 (the Brazilian National Survey on Child Nutrition) (n=5004).

| Variables | Mean or Frequency (%) | 95% CI |
|---|---|---|
| Region | | |
| North | 24.8 | 23.6; 26.0 |
| Midwest | 20.5 | 19.4; 21.7 |
| Southeast | 18.6 | 17.5; 19.7 |
| Northeast | 22.6 | 21.5; 23.8 |
| South | 13.5 | 12.6; 14.5 |
| Age (months) | 34.0 | 33.6; 34.5 |
| Age groups (months) | | |
| 6–23 | 29.6 | 28.3; 30.8 |
| 24–35 | 21.5 | 20.4; 22.7 |
| 36–59 | 48.9 | 47.5; 50.3 |
| Sex | | |
| Male | 51.1 | 49.7; 52.5 |
| Female | 48.9 | 47.5; 50.3 |
| Weight for length/height (z-score) | 0.3 | 0.29; 0.36 |
| Weight for length/height categories *,† | | |
| Underweight (z-score < –2) | 1.6 | 1.3; 2.0 |
| Normal (–2≤z-score≤1) | 73.3 | 72.1; 74.5 |
| Overweight risk (1<z-score≤2) | 17.3 | 16.3; 18.4 |
| Excessive weight (z-score >2) | 7.3 | 6.6; 8.0 |
| Mother/caregiver education (years) | | |
| 0–7 | 25.5 | 24.4; 26.8 |
| 8–10 | 23.5 | 22.3; 24.7 |
| ≥11 | 51.0 | 49.6; 52.4 |
| Mode of delivery | | |
| Vaginal | 53.7 | 52.3; 55.1 |
| Elective c-section | 19.0 | 18.0; 20.2 |
| Non-elective c-section | 27.3 | 26.1; 28.5 |
| Monthly family income (USD) ‡ | | |
| <62.20 | 3.8 | 3.3; 4.4 |
| 62.20–124.40 | 4.5 | 4.0; 5.1 |
| 124.50–248.70 | 19.2 | 18.2; 20.3 |
| >248.70 | 72.4 | 71.2; 73.7 |
| Minimum dietary diversity (MDD) § | | |
| ≥5 food groups | 59.3 | 39.4; 42.1 |
| <5 food groups | 40.7 | 57.9; 60.6 |

Note: CI: Confidence interval; USD: United States dollar.

*25 missing values.

†Reference WHO child growth standards, 2006 (**WHO, 2006**).

‡Estimated from the Brazilian minimum wage (R$ 998.00) and converted to the USD exchange 26 rate (R$ 4.013 = USD 1) in 2019 (Brazilian minimum wage = USD 248.70).

§MDD: frequency of children who received ≥ 5 or < 5 out of eight food groups on the day before the interview, [food groups: (1) breast milk; (2) grains, roots and tubers; (3) beans, nuts and seeds; (4) dairy products; (5) flesh foods; (6) eggs; (7) vitamin A-rich fruits and vegetables; and (8) fruits and vegetables] (**WHO, 2021**; **UNICEF, 2017**; **Supplementary file 1**).

The DQ mean (95% CI) was 0.98 (0.97; 0.99). Overall, children had lower DQs if they were male ($p=2 \times 10^{-14}$), older ($p<2 \times 10^{-16}$), had lower weight for height ($p=4 \times 10^{-5}$), <5 food groups than the MDD ($p=3 \times 10^{-4}$), were from the northern region ($p=2 \times 10^{-14}$), had a lower monthly family income ($r=0.05$, $p=4 \times 10^{-4}$), and a mother/caregiver with fewer years of education ($p=10^{-15}$). Mode of delivery was not significantly associated with DQ ($p=0.724$; *Supplementary file 1a*).

Given the size of our sample, statistical power is not an issue in our analyses. Considering an alpha of 0.05 for a two-sided test, a sample size of 5000 has 95% power to detect a correlation of $r=0.05$ and an effect of f2=0.003 in a multiple regression model with 4 predictors. As an initial assessment of the zero-order associations between DQ and serum metabolites, Pearson's correlations were performed. This revealed 26 negative and 2 positive statistically significant associations (*Figure 3—figure supplement 1*). Two unknown anions, annotated by their accurate mass, relative migration time, and ionization mode as 117.0552:1.67:N and 135.0293:1.71:N presented positive correlations. These two anions were tentatively identified as 3-hydroxyvaleric acid ($r=0.05$, 95% CI [0.02; 0.08], adjusted-p=0.001) and erythronic acid ($r=0.04$, 95% CI [0.01; 0.07], adjusted-p=0.011; *Figure 3—figure supplement 1*).

The highest correlations were all negative and were for serum phenylacetylglutamine (PAG, $r=-0.16$, 95% CI [–0.18; –0.13], adjusted-p=$10^{-26}$), cresol sulfate (CS, $r=-0.15$, 95% CI [–0.18; –0.12], adjusted-p=$10^{-24}$), hippuric acid (HA, $r=-0.14$, 95% CI [–0.17; –0.11], adjusted-p=$2 \times 10^{-22}$), creatinine (Crtn, $r=-0.13$, 95% CI [–0.16; –0.1], adjusted-p=$5 \times 10^{-19}$), trimethylamine-*N*-oxide (TMAO, $r=-0.1$, 95% CI [–0.13; –0.07], adjusted-p=$2 \times 10^{-11}$), citrulline (Cit, $r=-0.09$, 95% CI [–0.12; –0.07], adjusted-p=$3 \times 10^{-10}$), deoxycarnitine or γ-butyrobetaine (dC0, $r=-0.09$, 95% CI [–0.12; –0.06], adjusted-p=$3 \times 10^{-9}$), and 3-methylhistidine (MeHis, $r=-0.07$, 95% CI [–0.1; –0.05], adjusted-p=$10^{-6}$).

The model with three components was used for parsimony and to avoid overfitting. The serum metabolites that had the highest loads on the components were the branched-chain amino acids, including leucine (Leu), isoleucine (Ile), and valine (Val) on component 1, the uremic toxins, CS and PAG on component 2 and betaine and amino acids, mainly glutamine (Gln) and asparagine (Asn) on component 3 (*Figure 3—figure supplement 2*). The three components accounted for 39.8% of the total metabolite variance *Figure 3—figure supplement 3* and 4.3% of the DQ variance (*Figure 3—figure supplement 4*). Twenty-eight serum metabolites had a VIP ≥1 (*Figure 3*). These metabolites were then entered into the multiple linear regressions adjusted for the child's diet quality (MDD), nutritional status (weight for length/height z-scores - w/h z-score), and age (months), which were the minimum adjustments indicated by the DAG as described in the statistical analysis section.

We found inverse associations of serum concentrations of CS ($\beta=-0.07$; adjusted-p <0.001), HA ($\beta=-0.06$; adjusted-p <0.001), PAG ($\beta=-0.06$; adjusted-p <0.001), and TMAO ($\beta=-0.05$; adjusted-p=0.002) with the DQ of children, which were also significant in the models described below (*Table 2*). Since the child's diet and metabolism may change as the child ages and as neurodevelopmental disorders occur more frequently in boys than in girls, interactions between the metabolites and child age (in months) and between metabolites and child sex were also tested to evaluate a possible modification of the effects by these variables in the models. Considering the interactions between serum metabolites and child age, we observed associations for Crtn (β-interaction=0.05; adjusted-p=0.003), HA (β-interaction=0.04; adjusted-p=0.041), MeHis (β-interaction=0.04; adjusted-p=0.018), PAG (β-interaction=0.04; adjusted-p=0.018), TMAO (β-interaction=0.05; adjusted-p=0.003), and Val (β-interaction=0.04; adjusted-p=0.039; *Table 2*).

Comparing children one standard deviation (SD) above the mean child age with those one standard deviation below (49 months vs. 19 months), we observed opposite directions for the association with DQ for serum Crtn (for children aged - 1 SD: β = - 0.05; p=0.01;+1 SD: $\beta=0.05$; p=0.02) and for MeHis (- 1 SD: β = - 0.04; p=0.04;+1 SD: $\beta=0.04$; p=0.03; *Figure 4*). For serum TMAO, PAG, Val, and HA, the effect size went from a negative value for younger children to a non-significant value for older children (*Figure 4*). No associations were found for interactions between child sex and each metabolite on DQ (data not shown).

Mediation analyses identified that serum PAG was a mediator for the relationship between mode of delivery (ACME = 0.003, $p=2 \times 10^{-16}$), child's diet quality (ACME = 0.002, p=0.019), and child fiber intake (ACME = - 0.002; p=0.034) and DQ (*Figure 3—figure supplement 5*). Serum HA (ACME = - 0.004, p<0.001) and TMAO (ACME = - 0.002, p=0.022) were also mediators for the relationship between child fiber intake and DQ (*Figure 3—figure supplement 5*). According to the mediation analysis, having a vaginal delivery ($\beta=-0.05$; p<0.001), not achieving MDD ($\beta=-0.03$; p=0.019), and

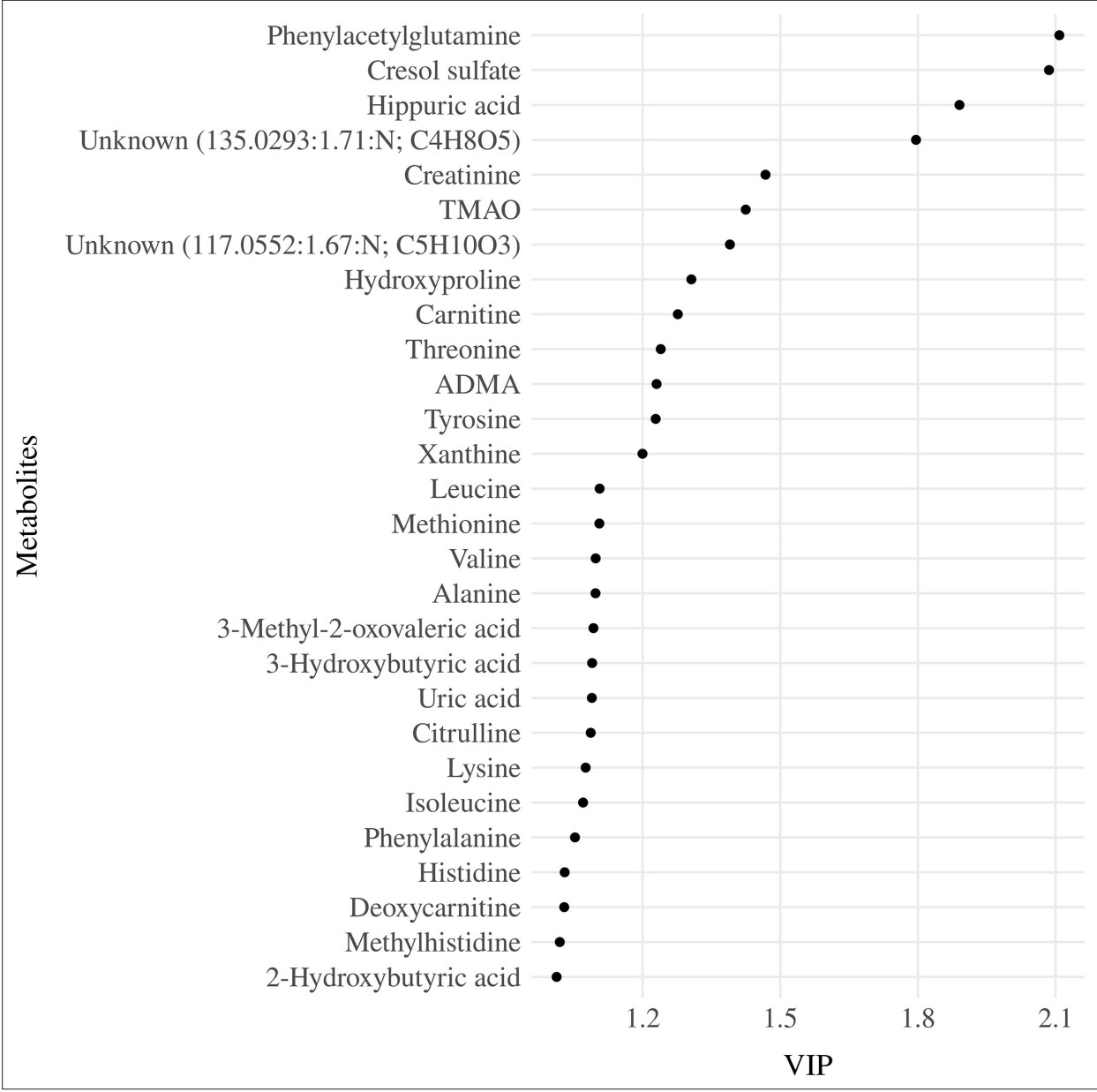

**Figure 3.** Variable importance projection (VIP) ranking scores from Partial Least Squares Regression (PLSR) analysis were performed to select the most relevant metabolites that explain the developmental quotient variability. Note: Serum metabolites with higher VIP scores are the most contributory variables, whereas smaller VIP scores provide less contribution to explain the developmental quotient variability. Only metabolites with VIP scores higher than one are displayed.

The online version of this article includes the following figure supplement(s) for figure 3:

**Figure supplement 1.** Volcano plot for Pearson correlations between serum metabolome and developmental quotients of children 6-59 months evaluated in ENANI-2019 (the Brazilian National Survey on Child Nutrition; n = 5004).

**Figure supplement 2.** Partial Least Square Regression (PLSR) loading plots for components 1, 2, and 3.

**Figure supplement 3.** Partial Least Square Regression (PLSR) score scatterplots for components 1, 2, and 3.

**Figure supplement 4.** Residual means squared error (RMSEP) and R squared across Partial Least Square Regression (PLSR) models with different numbers of components.

**Figure supplement 5.** Metabolite-mediated effects of mode of delivery, child's diet quality, and fiber Intake on child developmental quotient.

**Table 2.** Association between serum metabolites and developmental quotient (DQ) of children with and without child age interaction term for children 6–59 months evaluated in a subset sample of ENANI-2019 (the Brazilian National Survey on Child Nutrition; n=5004).

| Serum Metabolites | Main effect without child age interaction term* | | | Main effect with child age interaction term*, † | | | Interaction term (Metabolites: Child age)*, † | | |
|---|---|---|---|---|---|---|---|---|---|
| | β | 95% CI | Adjusted p-value | β | 95% CI | Adjusted p-value | β | 95% CI | Adjusted p-value |
| Asymmetric dimethylarginine | <0.01 | –0.03; 0.02 | 0.942 | <0.01 | –0.03; 0.02 | 0.984 | <0.01 | –0.02; 0.03 | 0.965 |
| Alanine | –0.02 | –0.04; 0.01 | 0.431 | –0.02 | –0.04; 0.01 | 0.414 | <0.01 | –0.03; 0.02 | 0.928 |
| Carnitine | <0.01 | –0.03; 0.03 | 0.954 | <0.01 | –0.03; 0.03 | 0.985 | <0.01 | –0.02; 0.03 | 0.965 |
| Citrulline | –0.03 | –0.05; 0.00 | 0.164 | –0.03 | –0.05; 0.00 | 0.176 | 0.02 | 0.00; 0.05 | 0.186 |
| Creatinine | –0.01 | –0.03; 0.02 | 0.935 | <0.01 | –0.03; 0.03 | 0.985 | 0.05 | 0.02; 0.08 | **0.003** |
| Cresol sulfate | –0.07 | –0.10; –0.04 | **<0.001** | –0.06 | –0.09; –0.03 | **<0.001** | 0.03 | 0.01; 0.06 | 0.059 |
| Deoxycarnitine | –0.03 | –0.06; –0.01 | 0.065 | –0.03 | –0.06; –0.01 | 0.073 | 0.02 | –0.01; 0.05 | 0.227 |
| Hippuric acid | –0.06 | –0.09; –0.04 | **<0.001** | –0.06 | –0.09; –0.03 | **<0.001** | 0.04 | 0.01; 0.06 | **0.041** |
| Histidine | –0.02 | –0.05; 0.00 | 0.183 | –0.02 | –0.05; 0.00 | 0.183 | 0.02 | 0.00; 0.05 | 0.156 |
| Hydroxyproline | –0.01 | –0.03; 0.02 | 0.912 | –0.01 | –0.03; 0.02 | 0.976 | –0.02 | –0.05; 0.01 | 0.227 |
| Isoleucine | –0.03 | –0.05; 0.00 | 0.164 | –0.03 | –0.05; 0.00 | 0.176 | 0.02 | –0.01; 0.04 | 0.312 |
| Leucine | –0.02 | –0.05; 0.00 | 0.183 | –0.02 | –0.05; 0.00 | 0.203 | 0.03 | 0.00; 0.05 | 0.144 |
| Lysine | <0.01 | –0.03; 0.02 | 0.935 | <0.01 | –0.03; 0.02 | 0.984 | 0.02 | 0.00; 0.05 | 0.194 |
| Methionine | –0.03 | –0.05; 0.00 | 0.164 | –0.03 | –0.05; 0.00 | 0.176 | 0.02 | –0.01; 0.05 | 0.227 |
| Methylhistidine | <0.01 | –0.02; 0.03 | 0.952 | <0.01 | –0.03; 0.03 | 0.985 | 0.04 | 0.02; 0.07 | **0.018** |
| Phenylacetylglutamine | –0.06 | –0.09; –0.04 | **<0.001** | –0.06 | –0.09; –0.03 | **0.001** | 0.04 | 0.01; 0.07 | **0.018** |
| Phenylalanine | –0.03 | –0.06; 0.00 | 0.164 | –0.03 | –0.05; 0.00 | 0.176 | 0.03 | 0.00; 0.05 | 0.129 |
| Threonine | <0.01 | –0.03; 0.02 | 0.953 | <0.01 | –0.03; 0.02 | 0.985 | 0.01 | –0.02; 0.03 | 0.836 |
| TMAO | –0.05 | –0.08; –0.02 | **0.002** | –0.04 | –0.07; –0.02 | **0.010** | 0.05 | 0.02; 0.07 | **0.003** |
| Tyrosine | –0.01 | –0.04; 0.02 | 0.816 | –0.01 | –0.04; 0.02 | 0.810 | 0.02 | 0.00; 0.05 | 0.194 |
| 117.0552:1.67:N; C5H10O3 | –0.02 | –0.05; 0.01 | 0.317 | –0.02 | –0.05; 0.01 | 0.322 | <0.01 | –0.03; 0.03 | 0.965 |
| 135.0293:1.71:N; C4H8O5 | 0.01 | –0.02; 0.04 | 0.721 | 0.01 | –0.02; 0.04 | 0.653 | –0.01 | –0.04; 0.01 | 0.524 |
| Uric acid | –0.01 | –0.03; 0.02 | 0.924 | <0.01 | –0.03; 0.02 | 0.984 | 0.02 | 0.00; 0.05 | 0.150 |
| Valine | –0.03 | –0.05; 0.00 | 0.164 | –0.02 | –0.05; 0.00 | 0.183 | 0.04 | 0.01; 0.06 | **0.039** |
| 2-Hydroxybutyric acid | –0.01 | –0.04; 0.02 | 0.800 | –0.01 | –0.04; 0.02 | 0.653 | 0.03 | 0.00; 0.06 | 0.129 |
| 3-Hydroxybutyric acid | –0.03 | –0.05; 0.00 | 0.164 | –0.03 | –0.05; 0.00 | 0.176 | <0.01 | –0.03; 0.03 | 0.965 |
| 3-Methyl-2-oxovaleric acid | <0.01 | –0.03; 0.03 | 0.957 | <0.01 | –0.03; 0.03 | 0.985 | <0.01 | –0.02; 0.03 | 0.965 |
| Xanthine | <0.01 | –0.02; 0.03 | 0.954 | <0.01 | –0.03; 0.03 | 0.985 | <0.01 | –0.03; 0.03 | 0.965 |

Note: CI: Confidence interval; DQ: Developmental quotient; TMAO: Trimethylamine N-oxide.

Only metabolites with variable importance projection (VIP) scores > 1 were entered in the regressions (29 serum metabolites). Adjusted p-values refer to p-values adjusted for 802 multiple comparisons by the Benjamini-Hochberg method. The child development was based on the Survey of Well-being of Young Children - the Brazilian version of the 803 milestones questionnaire, which estimates the child's developmental age considering developmental milestones achieved. The developmental quotient was calculated as the 804 child's developmental age divided by the chronological age (Sheldrick & Perrin, 2013). All variables were scaled before running the regression models. Bold values indicate 805 statistically significant adjusted p-values.

*Multiple linear regression adjusted for child's diet quality (minimum dietary diversity), child nutritional status (w/h score), and child age (in months).

†Child age (in months) was used as interaction in the regression models.

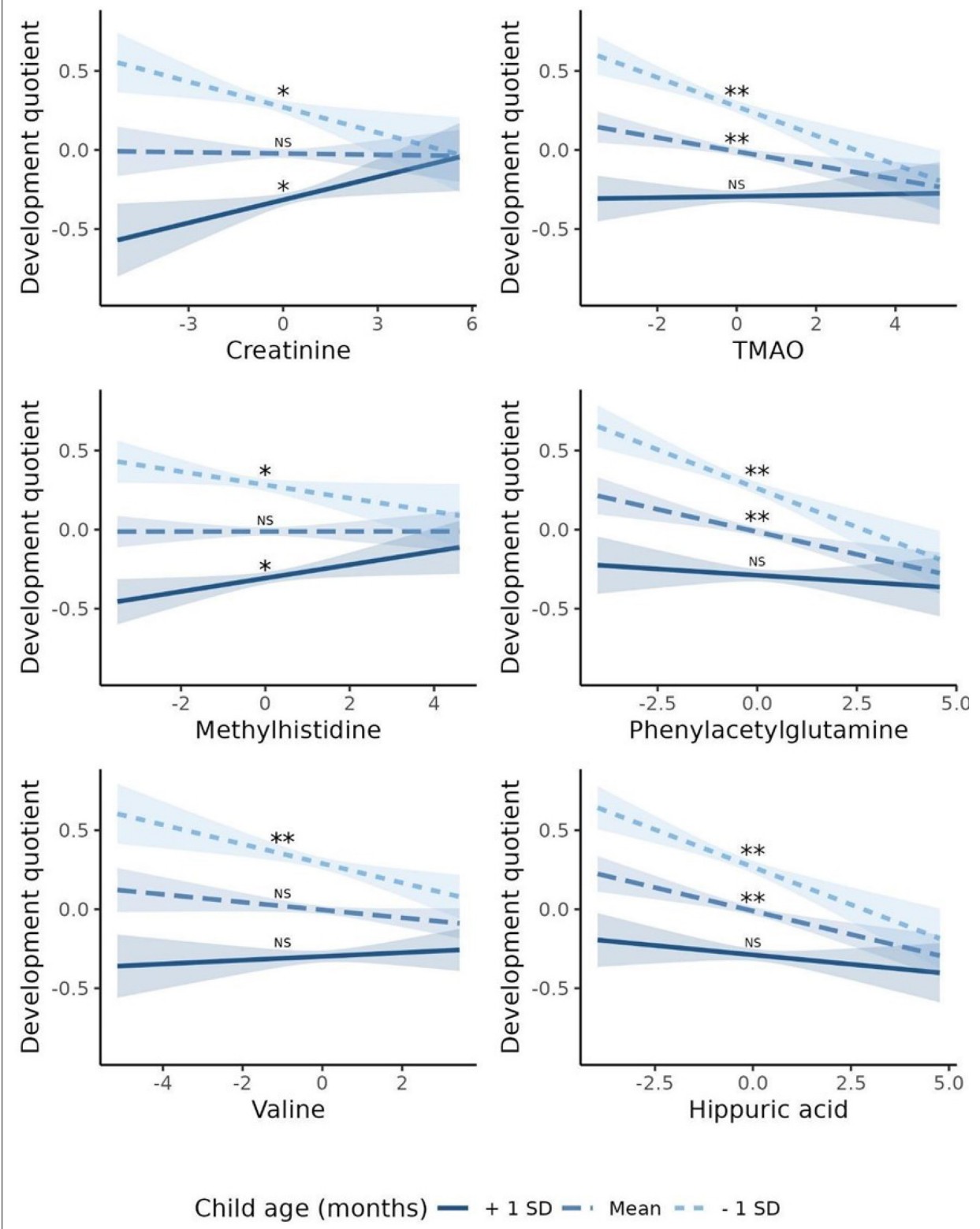

**Figure 4.** Association between serum metabolites with child age interaction and developmental quotient of children from 6 to 59 months from ENANI-2019 (the Brazilian National Survey on Child Nutrition, 2019) (n=5004). Note: SD: standard deviation; TMAO: Trimethylamine N-oxide. Only statistically significant interactions at an adjusted p-value for multiple comparisons <0.05 were decomposed into their simple slopes. ** refers to simple slopes with p-value <0.001; * refers to simple slopes with p-value <0.05; 'NS' refers to 'not statistically significant'. Regression adjustments used in the models: child diet quality, child age (months), and child nutritional status (w/h score). Mean of children age = 34 months and standard deviation

*Figure 4 continued*

= 15 months. The child development was based on the Survey of Well-being of Young Children - the Brazilian version of milestones questionnaire, which estimates the child's developmental age considering developmental milestones achieved. The developmental quotient (DQ) was calculated as the child's developmental age divided by the chronological age (*Sheldrick and Perrin, 2013*). All variables' data were auto-scaled, before running the regression models. *Figure 3—figure supplement 1*. Volcano plot for Pearson correlations between serum metabolome and developmental quotients of children 6-59 months evaluated in ENANI-2019 (the Brazilian National Survey on Child Nutrition; n = 5004). Note: The Y-axis and X-axis represent the negative logarithm (base-10) of adjusted p-values of each correlation coefficient and the correlation coefficient, respectively. Dashed line is the Benjamini-Hochberg adjusted-p-value for 72 metabolites comparisons. Statistically significant serum metabolites are labeled on plots.

greater total fiber intake ($\beta$=0.03; p=0.031) increased the serum PAG concentration, that in turn was inversely associated with DQ. Moreover, a higher dietary fiber intake was directly associated with HA ($\beta$=0.06; p<0.001) and TMAO ($\beta$=0.03; p=0.018), which also was inversely associated with DQ.

## Discussion

A limited number of investigations have examined the link between blood, urine, or stool metabolites and early stages of child development, with most studies focusing on comparing the metabolic profile of patients with developmental disorders against healthy controls (*Brydges et al., 2021*; *Moreau et al., 2019*; *Needham et al., 2021*; *Ruggeri et al., 2014*; *Orozco et al., 2019*). This is the first study to explore the association between child serum metabolome and ECD on a population-based level. According to our results, serum concentrations of PAG, CS, HA and TMAO were inversely associated with child DQ, a validated measure to express ECD. The associations of PAG, HA, TMAO, and Val on DQ were also age-dependent and showed stronger associations for children <34 months. In addition, inverse associations were found for serum levels of MeHis and Crtn with DQ for children <34 months, whereas direct associations were found for children >34 months.

PAG is the glutamine conjugate of phenylacetic acid generated from the gut microbial-dependent metabolism of phenylalanine (*Krishnamoorthy et al., 2024*; *Poesen et al., 2016*). As circulating concentrations of the essential amino acid, phenylalanine, were not directly associated with DQ in our study, this suggests that differences in gut microbiome composition impacting PAG formation among children are likely a major determinant of DQ rather than dietary protein intake. Similarly, to our findings, a previous study involving 76 patients with Attention-Deficit/Hyperactivity Disorder (ADHD) and 363 healthy children aged 1–18 years identified an inverse relationship between urinary PAG and ADHD (*Tian et al., 2022*). While the specific pathways contributing to such disorders remain to be fully elucidated, it is known that PAG is structurally similar to catecholamines and can activate adrenergic receptors (*Huynh, 2020*). The stimuli of adrenergic receptors may have broader implications on behavioral responses, potentially influencing neurological activities (*Connors et al., 2005*; *Pliszka et al., 1996*).

Likewise, in our study, circulating TMAO levels were inversely associated with child DQ. Elevated concentrations of TMAO in plasma and cerebrospinal fluid are also implicated in age-related cognitive dysfunction, neuronal senescence, and synaptic damage in the brain (*Praveenraj et al., 2022*). In addition, its increased levels have been associated with activation of inflammatory pathways (*Seldin et al., 2016*) and neurodegenerative diseases (*Mudimela et al., 2022*). Previous studies reported that TMAO can activate astrocytes and microglia and trigger a cascade of inflammatory responses in the brain, induce oxidative stress, superoxide production, and mitochondrial impairment, and cause inhibition of mTOR signaling in the brain (*Mudimela et al., 2022*; *Praveenraj et al., 2022*). In this context, dysregulation of the mTOR signaling pathway may lead to substantial abnormalities in brain development, contributing to a wide array of neurological disorders, including ASD, seizure, learning impairments, and intellectual disabilities (*Altomare and Gitto, 2015*; *Lee, 2015*).

HA is a glycine conjugate derived from exposure to benzoic acid (i.e. preservative in processed foods), or generated via intestinal microbial fermentation of dietary polyphenols and phenylalanine (*Assem et al., 2018*). Like other co-metabolized species, circulating HA concentrations depend on dietary exposures and host metabolism (*Jian et al., 2021*; *Khan et al., 2022*), in a case-control study involving 65 children with ASD and 20 children with typical development, reported that urinary HA was significantly higher in the ASD group, corroborating the inverse association found with DQ in our study. However, the effect of HA on metabolic health is still controversial as it has been proposed as

a potential dietary biomarker for fruit and vegetable consumption in healthy children and adolescents (*Krupp et al., 2012*; *Pallister et al., 2017*). HA also inhibits the Organic Anion Transporter (OAT) 3 function and contributes to the toxic action of other compounds, including indoxyl sulfate (*Ticinesi et al., 2023*), which may affect cognitive function by disrupting the brain barrier (*Lin et al., 2019*).

CS is a product of tyrosine fermentation in the gut involving more than 55 p-cresol producing bacteria prior to hepatic sulfate conjugation (*Saito et al., 2018*). CS was inversely associated with DQ in our study, and it has been studied in the early stages of life, particularly concerning conditions such as ASD (*Guzmán-Salas et al., 2022*; *Persico and Napolioni, 2013*). Urinary p-cresol and CS are elevated in ASD-diagnosed children <8 years (*Altieri et al., 2011*; *Persico and Napolioni, 2013*). Animal models have shown that CS is a gut-derived neurotoxin that can impact neuronal cell structural remodeling even at low doses via oxidative stress and secretion of brain-derived neurotrophic factor (*Tevzadze et al., 2022*). Indeed, p-cresol might impact developmental processes since it is related to impaired dendritic development, synaptogenesis, and synapse function in hippocampal neurons, which are crucial for cognitive and neural development in children (*Guzmán-Salas et al., 2022*).

Prior investigations have identified PAG, CS, HA, and TMAO as products of gut microbiota metabolism (*Pallister et al., 2017*; *Reichard et al., 2022*). Specifically, dietary aromatic amino acids are metabolized by gut microbiota in the large intestine, converting phenylalanine into PAG and HA and tyrosine into CS (*Reichard et al., 2022*; *Ticinesi et al., 2023*). In contrast, TMAO originates from trimethylamine (TMA), which is produced from betaine compounds, including γ-butyrobetaine (dC0), choline, and carnitine via gut microbiota co-metabolism that is subsequently oxidized to TMAO in the liver (*Reichard et al., 2022*).

These metabolic compounds in the bloodstream may elicit physiological responses influencing the central nervous system through direct passage across the blood-brain barrier or indirectly through vagus nerve stimulation (*Morais et al., 2021*). Such dynamics underscore the complex interactions between environmental exposures early in life, such as mode of birth and the child's diet, and brain development in early childhood (*Azad et al., 2018*).

Overall, serum PAG, HA, and TMAO showed a significant average causal mediation effect with dietary fiber intake that was inversely associated with DQ. However, the interpretation of mediation effects is limited by the observational nature of the data, and third variables may explain unexpected relationships between the variables in the analysis. Nevertheless, the results of our mediation analysis are an important step that future studies can build to further investigate the causal pathways leading to optimal DQ levels.

The age-dependent associations observed in our study are consistent with the age-related changes in metabolic profile reported in previous studies (*Chiu et al., 2016*; *Gu et al., 2009*; *Psihogios et al., 2008*; *Tian et al., 2022*). Increased urinary TMAO and betaine levels were found in children aged six months, whereas creatine and Crtn levels increased significantly after six months (*Chiu et al., 2016*). Similar findings were reported by Gu et al. in a study that included children from newborn to 12 years of age. The urinary Crtn increased with age, whereas glycine, betaine/TMAO, citrate, succinate, and acetone decreased (*Gu et al., 2009*). These changes may reflect a physiological age-dependent process related to the rapid growth occurring in early life (*Chiu et al., 2016*).

Interestingly, we observed that the child's age changed the direction of the association between Crtn and MeHis with DQ. Crtn is generated non-enzymatically from creatine and is related to energy production within skeletal muscle tissue, whereas MeHis is related to protein turnover and has been evaluated as a biomarker for the rate of skeletal muscle breakdown (*Kreider and Stout, 2021*; *Wang et al., 2012*). For example, plasma and urinary MeHis are temporally associated with changes to a health-promoting Prudent diet in contrast to a Western diet, whose concentrations are positively correlated with greater self-reported daily protein intake (*Wellington et al., 2019*). We hypothesize that higher serum concentrations of Crtn and MeHis in older children (>49 months) may be due to the greater physical activity/mobility needed through the first years (*Chiu et al., 2016*).

Our study provided valuable insights into the potential role of serum metabolome on ECD for children aged 6–59 months. One of the strengths of this study is the large sample size, which allows for a more comprehensive representation of the population on a national level. Using a subset of 5004 due to cost restrictions did not compromise the representativeness. The sample was randomly selected to constitute the final sample that aimed to represent the

total number of children with blood drawn (8829 children). Hence, our efforts were to preserve the original characteristics of the sample and the representativeness of the original sample. Furthermore, our study employed a quantitative targeted and exploratory untargeted metabolomics method. This high-throughput metabolomics platform is strengthened by implementing rigorous quality control measures and batch-correction algorithms, ensuring the high accuracy and reproducibility needed for large-scale epidemiological studies. We used the DQ as a variable for evaluating ECD, which consists of a continuous parameter that integrates developmental milestones attained with the child's chronological age at its achievement. DQ has been previously used (*Sheldrick et al., 2019* and *Freitas-Costa et al., 2023*) has advantages such as enabling the assessment of each item rather than just the final score, as the item set might be biased—meaning there could be an imbalance in the number of activities more commonly achieved among the specified items. Consequently, reaching the maximum score on the scale may be easier for certain age groups. Some limitations are worth mentioning. First, this study did not include the analysis of hydrophobic/water-insoluble lipids, limiting overall metabolome coverage. Also, the inherent limitations of a cross-sectional study prevent us from making causal inferences concerning the temporal relationship between serum metabolic phenotypes and ECD trajectories. Moreover, birth weight and breastfeeding practices were available only for a limited number of participants and were not included in the regression adjustments. Concerning the child's diet assessment, we estimated dietary diversity and fiber intake based on one1-day food intake reports, with the MDD specifically measuring dietary diversity within diet quality. Lastly, stool microbiome data was not collected from children in ENANI-2019 as it was not a study objective in this large population-based nutritional survey. However, the lack of microbiome data does not reduce the importance/relevance since there is no evidence that microbiome and factors affecting microbiome composition are confounders in the association between serum metabolome and child development.

In conclusion, this study represents a pioneering effort in Brazil, a population-based survey targeting children from 6 to 59 months of age that incorporated serum metabolome and ECD analysis. We found that serum PAG, HA, CS, and TMAO were inversely associated with ECD and that age can modify the effect of PAG, HA, TMAO, Crtn, and MeHis on development. Some circulating metabolites associated with DQ are relevant to microbiome health, it is possible that those circulating metabolites may be biomarkers of a beneficial effect of the microbiome on development. These results suggest that a panel of circulating metabolites might offer a preliminary warning of developmental risk and potentially be used as a screening tool to help identify children at risk for developmental delays at early stages of life.

We believe the results can contribute to the literature, which should be used to accumulate evidence to overcome knowledge gaps and support the formulation and redirection of public policies aimed at full child growth and development, the promotion of adequate and healthy nutrition and food security; the encouragement, support, and protection of breastfeeding; and the prevention and control of micronutrient deficiencies. Further prospective longitudinal studies, including stool-based microbiome analysis, are warranted to validate our findings and establish targeted intervention biomarkers besides providing further insights into possible mechanistic pathways.

## Acknowledgements

We thank the participating families who made this study possible and the Brazilian National Survey on Child Nutrition (ENANI-2019) team collaborators for their support in the fieldwork and organization of the database. We also thank the Brazilian Ministry of Health and Brazilian National Research Council (CNPq; process n. 440890/2017–9). PBM also acknowledges support from the Canada Foundation for Innovation and Genome Canada.

## Additional information

### Funding

| Funder | Grant reference number | Author |
|---|---|---|
| Brazilian National Research Council | | Meera Shanmuganathan |

The funders had no role in study design, data collection and interpretation, or the decision to submit the work for publication.

### Author contributions

Marina Padilha, Conceptualization, Data curation, Software, Formal analysis, Investigation, Visualization, Methodology, Writing – original draft; Victor Nahuel Keller, Data curation, Software, Formal analysis, Investigation, Visualization, Methodology, Writing – original draft; Paula Normando, Conceptualization, Investigation, Methodology, Writing – original draft, Project administration; Raquel M Schincaglia, Data curation, Formal analysis, Validation, Investigation, Methodology, Writing – original draft; Nathalia C Freitas-Costa, Data curation, Methodology, Writing – original draft; Samary SR Freire, Conceptualization, Data curation, Writing – review and editing; Felipe M Delpino, Formal analysis, Validation, Writing – original draft; Inês RR de Castro, Elisa MA Lacerda, Dayana R Farias, Conceptualization, Validation, Investigation, Visualization, Methodology, Writing – review and editing; Zachary Kroezen, Meera Shanmuganathan, Philip Britz-Mckibbin, Data curation, Formal analysis, Investigation, Visualization, Methodology, Writing – review and editing; Gilberto Kac, Conceptualization, Resources, Supervision, Funding acquisition, Validation, Investigation, Visualization, Methodology, Writing – original draft, Project administration, Writing – review and editing

### Author ORCIDs

Marina Padilha ⓘ https://orcid.org/0000-0002-1736-0411
Victor Nahuel Keller ⓘ https://orcid.org/0000-0002-4444-272X
Nathalia C Freitas-Costa ⓘ https://orcid.org/0000-0003-1798-0087
Samary SR Freire ⓘ https://orcid.org/0000-0001-7326-1058
Zachary Kroezen ⓘ https://orcid.org/0000-0001-9415-7604
Philip Britz-Mckibbin ⓘ https://orcid.org/0000-0001-9296-3223
Gilberto Kac ⓘ https://orcid.org/0000-0001-8603-9077

### Ethics

The ENANI-2019 was approved by the Research Ethics Committee of the Clementino Fraga Filho University Hospital of the Federal University of Rio de Janeiro (UFRJ) under the number CAAE 89798718.7.0000.5257. Data were collected after a parent/caregiver of the child authorized participation in the study through an informed consent form and following the principles of the Declaration of Helsinki.

### Decision letter and Author response

Decision letter https://doi.org/10.7554/eLife.97982.sa1
Author response https://doi.org/10.7554/eLife.97982.sa2

## Additional files

### Supplementary files

Supplementary file 1. (**a**) Means and confidence intervals (CI) for developmental quotient (DQ) by sociodemographic variables of children 6–59 months evaluated in a subset sample of ENANI-2019 (the Brazilian National Survey on Child Nutrition) (n=5004). (**b**) Annotation of unknown ions from untargeted metabolome analysis of serum filtrate samples by MSI-CE-MS from the ENANI-2019 study.

MDAR checklist

## Data availability

Data described in the manuscript, code book, and analytic code is available at: https://dataverse.nutricao.ufrj.br/dataverse/padilha-metab-dev.

The following dataset was generated:

| Author(s) | Year | Dataset title | Dataset URL | Database and Identifier |
|---|---|---|---|---|
| Keller VN | 2025 | All data and code | https://hdl.handle.net/20.500.12783/186 | Observatório de Epidemiologia Nutricional, 20.500.12783/186 |

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
