## [Editor Report]

This important work advances our understanding of elements influencing neurodevelopment in children. The data presented is convincing and offers insights into the effect of demographic and environmental factors, particularly nutrition, on the functioning of the gut-brain axis and the risk for developmental delays.

---

## [Decision Letter]

**Decision letter after peer review:**

Thank you for submitting your article "Serum metabolome indicators of early childhood development in the Brazilian National Survey on Child Nutrition (ENANI-2019)" for consideration by *eLife*. Your article has been reviewed by 3 peer reviewers, one of whom is a member of our Board of Reviewing Editors, and the evaluation has been overseen by a Senior Editor.

As is customary in *eLife*, the reviewers have discussed their critiques with one another and with the Reviewing and Senior Editors. The decision was reached by consensus. What follows below is the Reviewing Editor's edited compilation of the essential and ancillary points provided by reviewers in their critiques and in their interaction post-review. Please submit a revised version that addresses these concerns directly. Although we expect that you will address these comments in your response letter, we also need to see the corresponding revision clearly marked in the text of the manuscript. Some of the reviewers' comments may seem to be simple queries or challenges that do not prompt revisions to the text. Please keep in mind, however, that readers may have the same perspective as the reviewers. Therefore, it is essential that you amend or expand the text to clarify the narrative accordingly.

The Authors are invited to address the reviewer's comments, giving particular attention to data presentation/analyses, handling of confounding variables and additional support for aspects of the methodologies employed. Specifically, the Authors should consider the following points in their revision:

1) Inclusion of microbiome and factors affecting microbiome composition in the study as possible confounders

2) Possible shortcomings associated with the directed acyclic graph approach (clarifications as to the methodology employed; inclusion of the microbiome and additional factors affecting microbiome composition) and more broadly confounding variable selection

3) Lack of power calculations and sensitivity analyses to validate some of the strong assumptions made in subject selection, stratifications and covariate identifications

4) Relevance/appropriate use of PLSR

5) Discuss the use of DQ as a viable outcome

6) Biases in subject selection (towards higher socio-economic status) and generalizability of the study

*Reviewer #1 (Recommendations for the authors):*

I would recommend that the authors include all study covariates in Table 1, Supplementary Table 1 and the Methods section Covariates to prevent confusion. In addition to the mean and 95% CI, the p-values for DQ should also be included in Supplementary Table 1. All covariates that are statistically significantly associated with DQ in your study cohort should be included in the statistical models, and additionally studied via either interaction analysis or mediation analysis depending on which analysis is most appropriate. The directed acrylic graph (DAG) can be discarded completely. Following these suggestions should help prevent biasing the current analysis by the authors' perceived relevance and relationships of the covariates with the dependent and independent variables.

All significant results from the initial correlation analysis should be stated and considered, although the most striking ones on the volcano plot can be emphasized. Of benefit to the authors is that in the search for biomarkers, it matters less what causes the change in metabolite (e.g., diet, obesity) nor even if the metabolite has a direct effect on the neurodevelopmental outcome and is rather just a side effect of the true causative factor(s). If the concentration of a serum metabolite can reliably indicate the neurodevelopmental outcome, it has value as a biomarker regardless, and exploring the relationship with covariates and mechanism can be initially explored in this manuscript as you have done (i.e., interaction analysis and mediation analysis) and investigated in future work.

As part of the fuller description of the initial correlation analysis, a supplementary table should be included that provides more information on the unknown compounds, including the mass, confidence level of any identifications, likely chemical classes and/or chemical formulas.

If including all the significant covariates in the regression models is non-trivial (e.g., issues of multi-collinearity), the better use of PLSR may be done at this step to see if the variable importance of the metabolites exceeds the confounders, rather than its current use in metabolite selection which did not yield any findings beyond what was already found by the initial correlation analysis. Another option would be to use machine learning, such as random forest where false discovery rates can be estimated for importance metrics (R package pRF).

All abbreviations included in the manuscript should be defined at their first instance.

Although the DQ is an invaluable metric, it would also be of interest to explore the relationship between serum metabolite biomarkers and the different neurodevelopmental domains (e.g., motor, cognitive, language, socio-emotional) if possible.

*Reviewer #3 (Recommendations for the authors):*

1. Why was sensitivity analysis not done? Why was mediation analysis done and stopped? Shouldn't other testing have been done?

2. Though it is a large sample size, the sampling technique was convenience sampling. In resource-poor settings this may be the methodology that is followed but has funding been applied for? What are the plans for further study?

3. What was the time period of the data collection? Which year? Is it relevant in the world of 2024?

4. Can the data be compared with data from Brazil where the children were from lower socioeconomic strata or where mothers or caregivers education was lower or higher than what is mentioned in the study?

---

## [Author Response]

The Authors are invited to address the reviewer's comments, giving particular attention to data presentation/analyses, handling of confounding variables and additional support for aspects of the methodologies employed. Specifically, the Authors should consider the following points in their revision:1) Inclusion of microbiome and factors affecting microbiome composition in the study as possible confounders

Thank you for the comment and suggestions. It is important to highlight that there is no data on microbiome composition. We apologize if there was an impression such data is available. The main goal of conducting this national survey was to provide qualified and updated evidence on child nutrition to revise and propose new policies and nutritional guidelines for this demographic. Therefore, collection of stool derived microbiome (metagenomic) data was not one of the objectives of ENANI-2019. This is more explicitly stated as a study limitation in the revised manuscript on page 17, lines 463-467:

“Lastly, stool microbiome data was not collected from children in ENANI-2019 as it was not a study objective in this large population-based nutritional survey. However, the lack of microbiome data does not reduce the importance/relevance, since there is no evidence that microbiome and factors affecting microbiome composition are confounders in the association between serum metabolome and child development.”

Besides, one must consider the difficulties and costs in collecting and analyzing microbiome composition in a large population-based survey. In contrast, the metabolome data has been considered a priority as there was already blood specimens collected to inform policy on micronutrient deficiencies in Brazil. However, due to funding limitations we had to perform the analysis in a subset of our sample, still representative and large enough to test our hypothesis with adequate study power (more details below).

We would like to argue that there is no evidence that microbiome and factors affecting microbiome composition are confounders on the association between serum metabolome and child development. First, one should revisit the properties of a confounder according to the epidemiology literature that in short states that confounding refers to an alternative explanation for a given conclusion, thus constituting one of the main problems for causal inference (Kleinbaum, Kupper, and Morgenstern, 1991; Greenland and Robins, 1986; VanderWeele, 2019). In our study, we highlight that certain serum metabolites associated with the developmental quotient (DQ) in children were circulating metabolites (e.g., cresol sulfate, hippuric acid, phenylacetylglutamine, TMAO) previously reported to depend on dietary exposures, host metabolism and gut microbiota activity. Our discussion cites other published work, including animal models and observational studies, which have reported how these bioactive metabolites in circulation are co-metabolized by commensal gut microbiota, and may play a role in neurodevelopment and cognition as mediated by environmental exposures early in life.

In fact, the literature on the association between microbiome and infant development is very limited. We performed a search using terms ‘microbiome’ OR ‘microbiota’ AND ‘child development’ AND ‘systematic’ OR ‘meta-analysis’ and found only one study: ‘Associations between the human immune system and gut microbiome with neurodevelopment in the first 5 years of life: A systematic scoping review’ (DOI 10.1002/dev.22360). The authors conclude: ‘while the immune system and gut microbiome are thought to have interactive impacts on the developing brain, there remains a paucity of published studies that report biomarkers from both systems and associations with child development outcomes.’ It is important to highlight that our criteria to include confounders on the directed acyclic graph (DAG) was based on the literature of systematic reviews or meta-analysis and not on single isolated studies.

In summary, we would like to highlight that there is no microbiome data in ENANI-2019 and in the event such data was present, we are confident that based on the current stage of the literature, there is no evidence to consider such construct in the DAG, as this procedure recommends that only variables associated with the exposure and the outcome should be included. Please find more details on DAG below.

2) Possible shortcomings associated with the directed acyclic graph approach (clarifications as to the methodology employed; inclusion of the microbiome and additional factors affecting microbiome composition) and more broadly confounding variable selection

The use of DAGs has been widely explored as a valid tool for justifying the choice of confounding factors in regression models in epidemiology (Yan et al. 2024; Zhong et al. 2024; Al-Haddad et al. 2019; Chan et al. 2024). This is because DAGs allow for a clear visualization of causal relationships, clarify the complex relationships between exposure and outcome, and demonstrate the authors' transparency by acknowledging factors reported as important but not included/collected in the study. The statistical selection of variables, although classic, has become a less attractive approach as relies only on p-values and does not consider the biology and interaction between variables. This approach was a possibility, but the authors found the DAG is a more reliable and cutting-edge procedure. Besides, the DAG produced, and the suggested statistical adjustments performed to our regression models are very consistent.

Moreover, the DAG reduces the bias of data overfitting, which can lead to distortions in the results found. This error commonly occurs in other adjustment definition techniques, such as through bivariate statistical testing, since they do not guarantee causality, which is essential for causal inference. As described above, no evidence exists for the microbiome inclusion in DAG, which is otherwise not feasible to perform as stool specimens and metagenomic analyses were not collected/analyzed in ENANI-2019.

To make it clearer, we have modified the passage about DAG in the methods section. New text, page 9, lines 234-241:

“The subsequent step was to disentangle the selected metabolites from confounding variables. A Directed Acyclic Graph (DAG; Breitling et al., 2021) was used to more objectively determine the minimally sufficient adjustments for the regression models to account for potentially confounding variables while avoiding collider variables and variables in the metabolite-DQ causal pathways, which if controlled for would unnecessarily remove explained variance from the metabolites and hamper our ability to detect biomarkers. To minimize bias from subjective judgments of which variables should and should not be included as covariates, the DAG only included variables for which there was evidence from systematic reviews or meta-analysis of relationships with both the serum metabolome and DQ (Figure 1). Birth weight, breastfeeding, child's diet quality, the child's nutritional status, and the child's age were the minimal adjustments suggested by the DAG. Birth weight was a variable with high missing data, and indicators of breastfeeding practice data (referring to exclusive breastfeeding until 6 months and/or complemented until 2 years) were collected only for children aged 0–23 months. Therefore, those confounders were not included as adjustments. Child's diet quality was evaluated as MDD, the child's nutritional status as w/h z-score, and the child's age in months.”

3) Lack of power calculations and sensitivity analyses to validate some of the strong assumptions made in subject selection, stratifications and covariate identifications

To the best of our knowledge, there are no peer-reviewed manuscripts or books describing power calculation procedures for partial least squares (PLS)-based analyses. However, given our large sample size, we do not believe power was an issue with the analyses. For our regression analyses, which typically have 4 predictors, we had 95% power to detect an f-squared of 0.003 and an r of 0.05 in a two-sided correlation test considering an alpha level of 0.05.

Changes to the text were made to make this clearer to the reader. New text, page 11, lines 296-300:

“Given the size of our sample, statistical power is not an issue in our analyses. Considering an alpha of 0.05 for a two-sided test, a sample size of 5000 has 95% power to detect a correlation of r = 0.05 and an effect of f2 = 0.003 in a multiple regression model with 4 predictors. As an initial assessment of the zero-order associations between DQ and serum metabolites, Pearson’ s correlations were performed. This revealed 26 negative and 2 positive statistically significant associations (Figure 3—figure supplement 1).”

4) Relevance/appropriate use of PLSR

Thank you for the question. PLS-based analyses are among the most commonly used analyses for parsing metabolomic data (Blekherman et al., 2011; Wold et al., 2001; Gromski et al. 2015). This procedure is especially appropriate for cases in which there are multiple collinear predictor variables as it provides estimates of the metabolite predictiveness which are unbiased by multicollinearity between the metabolites, something a multiple linear regression analysis would be unable to achieve. PLSR is also more advantageous than testing each metabolite in separate analyses corrected for multiple comparisons (which would avoid bias from multicollinearity) because the correlated nature of the metabolites means the comparisons are not truly independent and would cause the corrections to be overly strict. Finally, the PLSR is more appropriate than the PLS-DA for our data because the dependent variable is continuous. Dichotomizing the dependent variable to be able to run a PLS-DA would entail a loss of information in the variable and less statistical power to detect relevant associations.

One of the reviewers questioned the use of PLSR on the basis that it merely ruled out some of the metabolites. Given that our goal is to select the most predictive serum metabolites in children from ENANI-2019, ruling out the less predictive metabolites is precisely what we aim to achieve. As explained above, the PLSR model allows us to reach that goal without introducing bias in our estimates or losing statistical power.

Considering this point, a passage on this aspect was added to the paper text.

New text, page 8, lines 220-224:

“To better assess the predictiveness of each metabolite in a single model, a PLSR was conducted. PLS-based analyses are the most commonly used analyses when determining the predictiveness of a large number of variables as they avoid issues with collinearity, sample size, and corrections for multiple-testing (Blekherman et al., 2011; Wold et al., 2001; Gromski et al. 2015).”

New text, page 12, lines 312-314:

“In PLSR analysis, the training data suggested that three components best predicted the data (the model with three components had the highest R2, and the root mean square error of prediction (RMSEP) was only slightly lower with four components). In comparison, the test data showed a slightly more predictive model with four components (Figure 3—figure supplement 2).”

5) Discuss the use of DQ as a viable outcome

Thank you for the suggestion. We are confident developmental quotient (DQ) is a viable outcome measure. The use of DQ has been discussed further in the manuscript to show is a consistent and useful measure of infant development.

In short, this method has been conceptualized by Sheldrick and Perrin (2013) and Sheldrick et al. (2019), the Survey of Well-being of Young Children creators. The goal was to provide an alternative for continuous assessments, considering that often times, development obeys a continuum, and its delay does not happen only when a threshold is reached. The DQ has been used previously by our team (Freitas-Costa et al. 2023) and other authors have used with different development tests, such as the Denver test (Drachler et al. 2007).

The DQ is a continuous variable calculated by dividing developmental age by chronological age. The developmental age was estimated using the item response theory according to the child's developmental milestones already reached. Differently for test scores use, these methods avoid the influence of items set in the results This method allows the evaluation of each responding item and not only the final score, as the item set can be biased—that is, there may be a disproportion in the number of activities more commonly achieved among the specified items. Therefore, reaching the maximum score on the scale may be easier in some age groups.This procedure allowed estimation in all the 12 age groups listed before, including milestones that did not apply and using chronological age as one of the estimators. The estimated model allowed the construction of an item characteristic curve (ICC) for each milestone, representing the change in the probability of a given response (sometimes or always) from birth to 62 months of age (age maximum in our data) and estimating the age of development.The final model generated ICC and 2 parameters (α and β) for each milestone. α1 and α2 refer to the probable age at which children reach each developmental milestone, performing them sometimes (α1) or always (α2), respectively. β refers to the discrimination of each milestone. It describes the curve slope, indicating the probability of reaching the developmental milestone, which increases with the child’s chronological age. ICC, α1, α2, and β made it possible to estimate developmental age according to the developmental milestones reached by each child.

Please find below a new paragraph that has been included in the Materials and methods section and new text in the Discussion section too with the main advantages of using DQ.

New text, page 16, lines 448-452:

“DQ has been previously used (Sheldrick et al., 2019 and Freitas-Costa et al., 2023) has advantages such as enabling the assessment of each item rather than just the final score, as the item set might be biased—meaning there could be an imbalance in the number of activities more commonly achieved among the specified items. Consequently, reaching the maximum score on the scale may be easier for certain age groups.”

New text, page 5-6, lines 134-152:

“The Developmental quotient (DQ) is a continuous variable calculated by dividing developmental age by chronological age. The item response theory and graded response models were used to estimate development age (Samejima, 1997) according to the child's developmental milestones already reached (Sheldrick et al. 2019). The analysis used the full information method and incorporated the complex sample design in the Mplus software version 7 (Los Angeles, EUA) (Muthén and Muthén, n.d.). The estimated model allowed the construction of an item characteristic curve (ICC) for each milestone, representing the change in the probability of a given response (sometimes or always) and the discrimination of each milestone development by age, estimating the development age (Freitas-Costa et al., 2023). The ICC and its coefficients were used to estimate developmental age according to the developmental milestones reached by each child. This methodology has been previously used to assess ECD with the SWYC (Freitas-Costa et al., 2023; Sheldrick et al., 2019; Sheldrick and Perrin, 2013) and the Denver Test (Drachler et al., 2007).

Differently for test scores use, these methods avoid the influence of items set in the results. This approach enables the assessment of each item rather than just the final score, as the item set might be biased—meaning there could be an imbalance in the number of activities that are more commonly achieved among the specified items. Consequently, reaching the maximum score on the scale may be easier for certain age groups. The developmental quotient (DQ) was calculated by dividing the developmental age by the chronological age (Freitas-Costa et al., 2023; Sheldrick and Perrin, 2013). DQ equals to 1 indicates that the expected age milestones are attained. DQ values < 1 or > 1 suggest attaining age milestones below or above expectations, respectively. This method allows analyzing the outcome as a continuous variable.”

6) Biases in subject selection (towards higher socio-economic status) and generalizability of the study

Thank you for highlighting this point. The ENANI-2019 is a population-based household survey with national coverage and representativeness for macroregions, sex, and one-year age groups (< 1; 1-1.99; 2-2.99; 3-3.99; 4-5). Furthermore, income quartiles of the census sector were used in the sampling. The study included 12,524 households 14,588 children, and 8,829 infants with blood drawn.

Due to the costs involved in metabolome analysis, it was necessary to further reduce the sample size to around 5,000 children that is equivalent to 57% of total participants from ENANI-2019 with stored blood specimens. To avoid a biased sample and keep the representativeness and generability, the 5,004 selected children were drawn from the total samples of 8,829 to keep the original distribution according age groups (6 to 11 months, 12 to 23 months, and 24 to 59 months), and some health conditions related to iron metabolism, e.g., anemia and nutrient deficiencies. Then, they were randomly selected to constitute the final sample that aimed to represent the total number of children with blood drawn. Hence, our efforts were to preserve the original characteristics of the sample and the representativeness of the original sample.

The ENANI-2019 study does not appear to present a bias towards higher socioeconomic status. Evidence from two major Brazilian population-based household surveys supports this claim. The 2017-18 Household Budget Survey (POF) reported an average monthly household income of 5,426.70 reais, while the Continuous National Household Sample Survey (PNAD) reported that in 2019, the nominal monthly per capita household income was 1,438.67 reais. In comparison, ENANI-2019 recorded a household income of 2,144.16 reais and a per capita income of 609.07 reais in infants with blood drawn, and 2,099.14 reais and 594.74 reais, respectively, in the serum metabolome analysis sample.

In terms of maternal education, the 2019 PNAD-Education survey indicated that 48.8% of individuals aged 25 or older had at least 11 years of schooling. When analyzing ENANI-2019 under the same metric, we found that 56.26% of ≥25 years-old mothers of infants with blood drawn had 11 years of education or more, and 51.66% in the metabolome analysis sample. Although these figures are slightly higher, they remain within a reasonable range for population studies.

It is well known that higher income and maternal education levels can influence child health outcomes, and acknowledging this, ENANI-2019 employed rigorous sampling methods to minimize selection biases. This included stratified and complex sampling designs to ensure that underrepresented groups were adequately included, reducing the risk of skewed conclusions. Therefore, the evidence strongly suggests that the ENANI-2019 sample is broadly representative of the Brazilian population in terms of both socioeconomic status and educational attainment.

Reviewer #1 (Recommendations for the authors):I would recommend that the authors include all study covariates in Table 1, Supplementary Table 1 and the Methods section Covariates to prevent confusion. In addition to the mean and 95% CI, the p-values for DQ should also be included in Supplementary Table 1. All covariates that are statistically significantly associated with DQ in your study cohort should be included in the statistical models, and additionally studied via either interaction analysis or mediation analysis depending on which analysis is most appropriate. The directed acrylic graph (DAG) can be discarded completely. Following these suggestions should help prevent biasing the current analysis by the authors' perceived relevance and relationships of the covariates with the dependent and independent variables.

To avoid further confusion, we have included all study covariates in Table 1 and calculated mean DQ estimates in Supplementary File 1(previously named Supplementary Table 1) for the same variables presented in Table 1. The relationships between DQ and the variables are described in the text (page 11) along with the appropriate test statistics and their p-values. For example, “Overall, children had lower DQs if they were male (p = 2 x 10^-14^), older (p < 2 x 10^-16^)…”

The reviewer recommendation to disregard the use of the DAG lacks a clear rationale and seems to be made because this is new to the reviewer or not used at all in his/her field. We would like to invite the reviewer to read a few seminal manuscripts about the procedure and consider other initiatives beyond the classic approaches (Greenland, Pearl and Robins, 1999; Hernán and Robins, 2020).

The use of DAGs for the selection of confounders of causal relationships is considered a cutting-edge procedure in the modern epidemiology literature (Yan et al. 2024; Zhong et al. 2024; Al-Haddad et al. 2019; Chan et al. 2024). Its advantages are exactly the opposite of what the reviewer says are its drawbacks. The DAG displays variables well documented considering systematic literature reviews and meta-analysis. Therefore, it does not include variables based on authors perceived relevance.

The DAG’s purpose is to reduce bias from subjective decisions on which variables to include and to select the appropriate covariates which are less likely to be colliders or to otherwise interfere with the estimates of the association between DQ and the metabolome. We describe this more explicitly in the article in page 28.

All significant results from the initial correlation analysis should be stated and considered, although the most striking ones on the volcano plot can be emphasized. Of benefit to the authors is that in the search for biomarkers, it matters less what causes the change in metabolite (e.g., diet, obesity) nor even if the metabolite has a direct effect on the neurodevelopmental outcome and is rather just a side effect of the true causative factor(s). If the concentration of a serum metabolite can reliably indicate the neurodevelopmental outcome, it has value as a biomarker regardless, and exploring the relationship with covariates and mechanism can be initially explored in this manuscript as you have done (i.e., interaction analysis and mediation analysis) and investigated in future work.

Thank you for the comments.

We agree that zero-order correlations between DQ and the metabolites are important in identifying biomarkers of DQ. However, when determining how reliable a biomarker is, it is also important to compare its predictive value with other correlated biomarkers (in the PLSR model) and consider whether it continues being a biomarker when other factors are taken into consideration (i.e., are controlled for in a regression) and for different levels of relevant variables (i.e., interaction effects). Therefore, we report zero-order correlations along with the PLSR and regression models to give a more complete picture of the usefulness of the metabolites as DQ biomarkers.

As part of the fuller description of the initial correlation analysis, a supplementary table should be included that provides more information on the unknown compounds, including the mass, confidence level of any identifications, likely chemical classes and/or chemical formulas.

Thank you very much for the suggestion, following your recommendation, we have added Supplementary File 2 with this information. Furthermore, we would like to clarify that all unknown ions reported in the ENANI-2019 study that were not identified with high confidence (level 1) by spiking with authentic standards and/or collision-induced dissociation experiments by MS/MS were annotated based on their accurate mass, relative migration time and ionization mode detected when using MSI-CE-MS. These unknown ions were also further authenticated after verifying that they were measurable with adequate precision in repeat pooled quality control (QC) samples throughput the study (mean CV < 40%), did not exhibit a signal in the blank (i.e., excluding background ions or artifact signals) and were also detected with adequate consistency in serum samples analyzed from the ENANI-2019 study (detection frequency > 70%). Furthermore, a tentative identification (level 3) was proposed for certain unknown ions after completing a search on the Human Metabolome Database (HMDB, https://hmdb.ca) based on viable candidates consistent with their charge state or functional group (e.g., basic amino functional group for cation or carboxylic acid functional group for anion) with reported concentrations in blood that were likely measurable (above low micromolar levels), whereas in other cases no candidate was reported. Further details on our data workflow for rigorous selection of molecular features when using performing untargeted serum metabolomic analyses as well as criteria used for metabolite annotation and identification is described elsewhere (Shanmuganathan et al. Nat. Protocol 2021, 16: 1966-1994).

If including all the significant covariates in the regression models is non-trivial (e.g., issues of multi-collinearity), the better use of PLSR may be done at this step to see if the variable importance of the metabolites exceeds the confounders, rather than its current use in metabolite selection which did not yield any findings beyond what was already found by the initial correlation analysis. Another option would be to use machine learning, such as random forest where false discovery rates can be estimated for importance metrics (R package pRF).

In our response to item #5 above we explain the function of PLSR and why it is the most appropriate multivariate model for the data and research questions we have. Because our study is concerned with serum metabolomic biomarkers of DQ, we focused on them instead of other covariates. Nevertheless, we ran the PLSR with the same covariates as in the regression models and found that, except for age (which is more strongly associated with DQ, as reported in the manuscript, r = 0.31), the covariates were less predictive of DQ than the majority of the metabolites (VIPs for age = 3.9, height/weight = 0.7, dietary diversity = 0.7). However, these results are difficult to interpret; we are unsure how the differing scales and distributions of the covariates, when compared to the metabolite data, affect their interpretation. Furthermore, the research we are familiar with does not include covariates in PLS analyses, so we believe it is safer to not include them.

All abbreviations included in the manuscript should be defined at their first instance.

All abbreviations have been defined at their initial appearance.

Although the DQ is an invaluable metric, it would also be of interest to explore the relationship between serum metabolite biomarkers and the different neurodevelopmental domains (e.g., motor, cognitive, language, socio-emotional) if possible.

It would be very interesting to explore the relationship between metabolomics and the different domains of child development. However, the SWCY is a screening tool that assesses the motor, language, and cognitive domains all together. Therefore, the result concerns the global/overall development. It is unfortunate but is not possible to distinguish which domain is at risk of delay. We chose this questionnaire because it is freely accessible, quick, easy to apply and can be used as a continuous outcome. In addition, it has been translated and cross-culturally adapted for use with the Brazilian population (Moreira et al., 2019). Therefore, it is feasible to be used in a population-based household survey conducted in a country with continental dimensions, such as Brazil. In the methods section and 'Assessment of ECD' subsection (page 5, lines 113-131), we describe the details of the SWYC and the domains it assesses.

Reviewer #3 (Recommendations for the authors):1. Why was sensitivity analysis not done? Why was mediation analysis done and stopped? Shouldn't other testing have been done?

We thank the reviewer for the question. It is unclear to us which sensitivity analysis the reviewer would have liked us to perform. We tested the association of serum metabolites and DQ under a number of different conditions (zero-order correlations, regressions with potential confounders, interactions with sex and age) and reported which metabolites held up across the analyses. In our view, this functions as a set of sensitivity tests for the metabolite-DQ relationship.

Regarding the mediation analyses, we noticed that our theoretical justifications for running it were in the statistical analysis section rather than the Results section. Therefore, our reasoning for running it was not clear when we presented the results. We have the new text from the statistical analysis section (page 9, lines 250-255):

“Since several circulating metabolites most associated with DQ are relevant to microbiome health, those circulating metabolites may be biomarkers of a beneficial effect of the microbiome on development. We employed mediation analyses to explore the potential role of specific serum metabolites as mediators in the relationship between certain exposure variables related to the microbiome establishment in early life, such as mode of delivery (Reyman et al., 2019), child's diet quality (Baldeon et al., 2023), as well as child fiber intake (Cronin et al., 2021) and DQ.”

2. Though it is a large sample size, the sampling technique was convenience sampling. In resource-poor settings this may be the methodology that is followed but has funding been applied for? What are the plans for further study?

The ENANI-2019 is a complex sampling household-based population survey on food and nutrition in children under five years of age. The sample was calculated at 15,000 households distributed across 123 municipalities (counties) in Brazil’s 26 states and the Federal District. The sample size, 3,000 households in each of the five major geographic regions, was defined to estimate a minimum proportion of 2% with a relative error of 35% (representing a range from 1.3% to 2.7%), 95% confidence level, and sampling design effect set at 2.

Brazil’s 5,570 municipalities, stratified in state capitals, large municipalities (population > 500,000), and other municipalities, were grouped by the major geographic regions. All the state capitals and large municipalities were included in the sample. In contrast, in the other strata, the municipalities were selected with probability proportional to the estimated number of children under five years of age on July 1, 2016. In the sampled municipalities, the census tracts were stratified by mean household income quartiles and selected with probability proportional to the number of children under five years of age, according to the 2010 Population Census, using Pareto sampling. The selection of households used an inverse sampling strategy until 10 households had been obtained in the tract or all the addresses in the tract had been visited, with any number of households interviewed. Details on the sampling plan are available in Vasconcellos MTL, Silva PLN, Castro IRR, Boccolini CS, Alves-Santos NH, Kac G. Sampling plan of the Brazilian National Survey on Child Nutrition (ENANI-2019): a population-based household survey. Cad Saúde Pública 2021; 37:e00037221.

Therefore, the sampling technique is far from being for convenience.

The 2019’s edition was funded by the Brazilian Ministry of Health, and in March 2024, a new edition of ENANI (ENANI-2024) commenced, again funded by the same agency (https://enani.nutricao.ufrj.br/).

3. What was the time period of the data collection? Which year? Is it relevant in the world of 2024?

As mentioned in Materials and methods – Study design and participants – ENANI-2019 data collection took place from February 2019 and ended in March 2020. Although time has passed since data was collected, we are confident that the results of this manuscript are relevant because the study aims to present the associations between children’s serum metabolome and early child development (ECD), a topic with limited literature and of great relevance. The presence or absence of this association is independent of the time or sociodemographic and epidemiological context, given that the association is intrinsic to metabolism and infant development characteristics. However, we agree that sociodemographic and epidemiological contexts may influence the metabolome and ECD, and these factors may have changed since the data collection.

4. Can the data be compared with data from Brazil where the children were from lower socioeconomic strata or where mothers or caregivers education was lower or higher than what is mentioned in the study?

Thank you for your comment. This has been already discussed in reply #6 to the reviewer #2 that has been pasted below for convenience.

Thank you for highlighting this point. The ENANI-2019 is a population-based household survey with national coverage and representativeness for macroregions, sex, and one-year age groups (< 1; 1-1.99; 2-2.99; 3-3.99; 4-5). Furthermore, income quartiles of the census sector were used in the sampling. The study included 12,524 households 14,588 children, and 8,829 infants with blood drawn.

Due to the costs involved in metabolome analysis, it was necessary to further reduce the sample size to around 5,000 children that is equivalent to 57% of total participants from ENANI-2019 with stored blood specimens. To avoid a biased sample and keep the representativeness and generability, the 5,004 selected children were drawn from the total samples of 8,829 to keep the original distribution according age groups (6 to 11 months, 12 to 23 months, and 24 to 59 months), and some health conditions related to iron metabolism, e.g., anemia and nutrient deficiencies. Then, they were randomly selected to constitute the final sample that aimed to represent the total number of children with blood drawn. Hence, our efforts were to preserve the original characteristics of the sample and the representativeness of the original sample.

The ENANI-2019 study does not appear to present a bias towards higher socioeconomic status. Evidence from two major Brazilian population-based household surveys supports this claim. The 2017-18 Household Budget Survey (POF) reported an average monthly household income of 5,426.70 reais, while the Continuous National Household Sample Survey (PNAD) reported that in 2019, the nominal monthly per capita household income was 1,438.67 reais. In comparison, ENANI-2019 recorded a household income of 2,144.16 reais and a per capita income of 609.07 reais in infants with blood drawn, and 2,099.14 reais and 594.74 reais, respectively, in the serum metabolome analysis sample.

In terms of maternal education, the 2019 PNAD-Education survey indicated that 48.8% of individuals aged 25 or older had at least 11 years of schooling. When analyzing ENANI-2019 under the same metric, we found that 56.26% of ≥25 years-old mothers of infants with blood drawn had 11 years of education or more, and 51.66% in the metabolome analysis sample. Although these figures are slightly higher, they remain within a reasonable range for population studies.

It is well known that higher income and maternal education levels can influence child health outcomes, and acknowledging this, ENANI-2019 employed rigorous sampling methods to minimize selection biases. This included stratified and complex sampling designs to ensure that underrepresented groups were adequately included, reducing the risk of skewed conclusions. Therefore, the evidence strongly suggests that the ENANI-2019 sample is broadly representative of the Brazilian population in terms of both socioeconomic status and educational attainment.

References used on the rebuttal letter

Blekherman, G., Laubenbacher, R., Cortes, D. F., Mendes, P., Torti, F. M., Akman, S.,.… and Shulaev, V. (2011). Bioinformatics tools for cancer metabolomics. Metabolomics, 7, 329-343.

Gromski, P. S., Muhamadali, H., Ellis, D. I., Xu, Y., Correa, E., Turner, M. L., and Goodacre, R. (2015). A tutorial review: Metabolomics and partial least squares-discriminant analysis–a marriage of convenience or a shotgun wedding. Analytica chimica acta, 879, 10-23.

Wold, S., Sjöström, M., and Eriksson, L. (2001). PLS-regression: a basic tool of chemometrics. Chemometrics and intelligent laboratory systems, 58(2), 109-130.

LUIZ, RR., and STRUCHINER, CJ. Inferência causal em epidemiologia: o modelo de respostas potenciais [online]. Rio de Janeiro: Editora FIOCRUZ, 2002. 112 p. ISBN 85-7541-010-5. Available from SciELO Books <http://books.scielo.org>.

GREENLAND, S. and ROBINS, J. M. Identifiability, exchangeability, and epidemiological Confounding. International Journal of Epidemiolgy, 15(3):413-419, 1986.

Freitas-Costa NC, Andrade PG, Normando P, et al. Association of development quotient with nutritional status of vitamins B6, B12, and folate in 6–59-month-old children: Results from the Brazilian National Survey on Child Nutrition (ENANI-2019). The American journal of clinical nutrition 2023;118(1):162-73. doi: https://doi.org/10.1016/j.ajcnut.2023.04.026

Sheldrick RC, Schlichting LE, Berger B, et al. Establishing New Norms for Developmental Milestones. Pediatrics 2019;144(6) doi: 10.1542/peds.2019-0374 [published Online First: 2019/11/16]

Drachler Mde L, Marshall T, de Carvalho Leite JC. A continuous-scale measure of child development for population-based epidemiological surveys: a preliminary study using Item Response Theory for the Denver Test. Paediatric and perinatal epidemiology 2007;21(2):138-53. doi: 10.1111/j.1365-3016.2007.00787.x [published Online First: 2007/02/17]

VanderWeele, TJ Princípios de seleção de fatores de confusão. Eur J Epidemiol 34, 211–219 (2019). https://doi.org/10.1007/s10654-019-00494-6

David G. Kleinbaum, Lawrence L. Kupper; Hal Morgenstern. Epidemiologic Research: Principles and Quantitative Methods. 1991

Yan R, Liu X, Xue R, Duan X, Li L, He X, Cui F, Zhao J. Association between internet exclusion and depressive symptoms among older adults: panel data analysis of five longitudinal cohort studies. EClinicalMedicine 2024;75. doi: 10.1016/j.eclinm.2024.102767.

Zhong Y, Lu H, Jiang Y, Rong M, Zhang X, Liabsuetrakul T. Effect of homemade peanut oil consumption during pregnancy on low birth weight and preterm birth outcomes: a cohort study in Southwestern China. Glob Health Action. 2024 Dec 31;17(1):2336312.

Aristizábal LYG, Rocha PRH, Confortin SC, et al. Association between neonatal near miss and infant development: the Ribeirão Preto and São Luís birth cohorts (BRISA). BMC Pediatr. 2023;23(1):125. Published 2023 Mar 18. doi:10.1186/s12887-023-03897-3

Al-Haddad BJS, Jacobsson B, Chabra S, et al. Long-term risk of neuropsychiatric disease after exposure to infection in utero. JAMA Psychiatry. 2019;76(6):594-602. doi:10.1001/jamapsychiatry.2019.0029

Chan, A.Y.L., Gao, L., Hsieh, M.HC. et al. Maternal diabetes and risk of attention-deficit/hyperactivity disorder in offspring in a multinational cohort of 3.6 million mother–child pairs. Nat Med 30, 1416–1423 (2024).

Hernan MA, Robins JM (2020). Causal Inference: What If. Boca Raton: Chapman and Hall/CRC.

Greenland S; Pearl J; Robins JM. Confounding and collapsibility in causal inference. Statist Sci. 14 (1) 29 – 46 1999. https://doi.org/10.1214/ss/1009211805